# Preventing Model Collapse Under Overparametrization: Optimal Mixing Ratios for Interpolation Learning and Ridge Regression

**Anvit Garg**[1], **Sohom Bhattacharya**[2] **& Pragya Sur**[1]*
[1]Harvard University, Cambridge, MA 02138
[2]University of Florida, Gainesville, FL 32611

## Abstract

Model collapse occurs when generative models degrade after repeatedly training on their own synthetic outputs. We study this effect in overparameterized linear regression in a setting where each iteration mixes fresh real labels with synthetic labels drawn from the model fitted in the previous iteration. We derive precise generalization error formulae for minimum-$\ell_2$-norm interpolation and ridge regression under this iterative scheme. Our analysis reveals intriguing properties of the optimal mixing weight that minimizes long-term prediction error and provably prevents model collapse. For instance, in the case of min-$\ell_2$-norm interpolation, we establish that the optimal real-data proportion converges to the reciprocal of the golden ratio for fairly general classes of covariate distributions. Previously, this property was known only for ordinary least squares, and additionally in low dimensions. For ridge regression, we further analyze two popular model classes – the random-effects model and the spiked covariance model – demonstrating how spectral geometry governs optimal weighting. In both cases, as well as for isotropic features, we uncover that the optimal mixing ratio should be at least one-half, reflecting the necessity of favoring real-data over synthetic. We study three additional settings: (i) where real data is fixed and fresh labels are not obtained at each iteration, (ii) where covariates vary across iterations but fresh real labels are available each time, and (iii) where covariates vary with time but only a fraction of them receive fresh real labels at each iteration. Across these diverse settings, we characterize when model collapse is inevitable and when synthetic data improves learning. We validate our theoretical results with extensive simulations.

## 1 Introduction

Modern AI models are increasingly trained on their own synthetic outputs. However, this practice can lead to *model collapse*, where prediction performance degrades progressively with iterative re-training on AI generated *synthetic data* Shumailov et al. (2024; 2023). The phenomenon has been empirically observed across a wide array of settings Alemohammad et al. (2024); Bohacek & Farid (2023); Briesch et al. (2023); Hataya et al. (2023); Martínez et al. (2023a;b). For instance, Alemohammad et al. (2024); Shumailov et al. (2024; 2023); Martínez et al. (2023a;b) studied settings where a model is iteratively trained on synthetic data generated from the previously trained model, starting from original real data. They also studied settings where a small fraction of the original real data is used in conjunction with the generated synthetic data. They demonstrated that model collapse occurs in both cases. These concerning observations motivated a surge in recent work that studies model collapse rigorously and attempts to develop method that mitigate it where possible Bertrand et al. (2024); Shumailov et al. (2024); Dohmatob et al. (2024a;b; 2025); Feng et al. (2025); Dey & Donoho (2024); Gerstgrasser et al. (2024); Kazdan et al. (2024). However, previous work remains limited to low-dimensional settings or Gaussian features, creating a significant gap in our understanding. This paper breaks this barrier by addressing two fundamental questions: Can mixing fresh real data with synthetic outputs mitigate model collapse in overparametrized problems? What

---

*Corresponding author: `pragya@fas.harvard.edu`

is the optimal mixing ratio that minimizes prediction error in the long run? We provide rigorous answers for overparametrized linear regression, demonstrating how model collapse can be prevented under overparametrization.

Prior work has rigorously studied model collapse across several problem settings, though with important limitations. For low-dimensional Gaussian distribution estimation, Shumailov et al. (2024) shows how repeated use of synthetic data causes the estimated covariance matrix to collapse to zero almost surely, while the sample mean diverges. Crucially, they consider settings where the original real data is fixed and fresh real data is unavailable at each iteration. Similar results hold for linear regression for Gaussian features Dohmatob et al. (2024a). Recent work Dohmatob et al. (2024b; 2025); Feng et al. (2025) attributes collapse to a change in scaling laws, with applications to text generation and Gaussian mixture problems. To mitigate model collapse, Bertrand et al. (2024); Gerstgrasser et al. (2024); Dey & Donoho (2024); Kazdan et al. (2024); He et al. (2025) develop a mixing framework, where models are trained on a mixture of real and synthetic data at each iteration. This approach prevents collapse by ensuring that the test error remains bounded even as the number of iterations increase. Crucially, these papers focus exclusively on low-dimensional problems.

We study two broad classes of estimators—minimum-$\ell_2$-norm interpolators (Section 3.1) and ridge regression (Section 3.2 and Section 3.3). Modern machine learning (ML) algorithms frequently exhibit implicit regularization Zhang & Yu (2005); Soudry et al. (2018); Gunasekar et al. (2018a)—with appropriate initialization and step sizes, algorithms converge to predictors that achieve remarkable generalization in overparameterized regimes. Implicit regularization has become a cornerstone for understanding why overparameterized models generalize well Bartlett et al. (2020). Within this framework, min-norm interpolators have emerged as a fundamental class of predictors that commonly arise as implicitly regularized limits of gradient-based algorithms Bartlett et al. (2020); Deng et al. (2022); Gunasekar et al. (2018a;b); Liang & Sur (2022); Montanari et al. (2019); Muthukumar et al. (2020); Soudry et al. (2018); Zhang & Yu (2005); Wang et al. (2022); Zhou et al. (2022). At the same time, ridge regression represents a fundamental learning paradigm that has historically provided valuable insights into complex algorithms, often illuminating phenomena observed in deep networks Hastie et al. (2022); Patil et al. (2024). We study these popular classes of predictors for understanding model collapse under overparametrization.

Our work here is motivated by findings in Alemohammad et al. (2024), which suggest that among interesting possible combinations of real and synthetic data mixing, fresh data augmentation is the only scenario where model collapse can be prevented. We work under this *fresh data augmentation* framework, which we describe below. This framework also appeared in He et al. (2025), though they focused on low-dimensional problems. Given covariates $\mathbf{X}$, at iteration $t$, we generate a new batch of *real* responses $\mathbf{y}_t$ alongside *synthetic* responses $\widetilde{\mathbf{y}}_t$ produced using the estimator from iteration $(t-1)$. At iteration $t$, the estimator is formed using a weighted mixture of these real and synthetic responses. Concretely, we define

$$\widehat{\boldsymbol{\beta}}_t = (\mathbf{X}^\top \mathbf{X})^\dagger \mathbf{X}^\top \big( w\,\mathbf{y}_t + (1-w)\,\widetilde{\mathbf{y}}_t \big).$$
$$\widehat{\boldsymbol{\beta}}_{t,\lambda} = (\mathbf{X}^\top \mathbf{X} + n\lambda\boldsymbol{I})^{-1}\mathbf{X}^\top \big( w\,\mathbf{y}_t + (1-w)\,\widetilde{\mathbf{y}}_{t,\lambda} \big). \tag{1.1}$$

Above, $\dagger$ denotes the pseudoinverse. It is well-known that $\widehat{\boldsymbol{\beta}}_t = \lim_{\lambda \to 0^+} \widehat{\boldsymbol{\beta}}_{t,\lambda}$. In the aforementioned setting, our main contributions are as follows:

(i) **Quantifying the generalization error.** In an overparametrized regime (stated precisely in Section 2), we characterize the generalization error as $t \to \infty$ for both the min-$\ell_2$-norm interpolator $\widehat{\boldsymbol{\beta}}_t$ (Theorem 3.1)and the ridge estimator $\widehat{\boldsymbol{\beta}}_{t,\lambda}$ (Theorem 3.4). Our results capture the precise dependence of the limiting risk on key problem parameters: the signal strength, feature covariance matrix, regularization level $\lambda$, and mixing proportion $w$. Our work substantially advances the growing literature on interpolation learning and high-dimensional ridge regression Montanari et al. (2019); Deng et al. (2022); Liang & Sur (2022); Wang et al. (2022); Zhou et al. (2022); Bach (2024); Patil et al. (2024); Mallinar et al. (2024); Song et al. (2024) (see also data selection strategies Rezaei et al. (2025)), which has previously not examined the impact of synthetic data on these learning problems.

(ii) **Characterizing the optimal mixing ratio.** We characterize the optimal weight on real labels that minimizes the long-term prediction error across different settings. For min-$\ell_2$-norm interpolation, we establish that the asymptotic risk is uniquely minimized at $w^\star = 1/\varphi$ (the reciprocal of the golden ratio) for *any* feature covariance matrix with bounded eigenvalues (see equation (2.3)). This

phenomenon was previously proved only for maximum likelihood estimators in generalized linear models and nonparametric estimation in He et al. (2025). But the focus here was on low-dimensional settings. For ridge regression, we prove that the risk is log-convex and admits a unique minimum at $w^\star$ under several important scenarios: when the covariance is isotropic (Theorem 3.3), or when the covariance follows a spiked model (Section 3.4.2), or when the signal follows a random-effects model (Theorem 3.2). Across all settings, we show $w^\star \geq 1/2$, highlighting the necessity of weighting real-data more heavily than synthetic data. Such rigorous analysis of the mixing ratio provides concrete guidance for mitigating model collapse in overparametrized problems, complementing recent theoretical studies that were limited to low-dimensional regression Gerstgrasser et al. (2024); Dey & Donoho (2024).

**Paper Structure**   The rest of the paper is structured as follows. Section 2 formalizes the problem setup and data-augmentation framework. In Section 3, we state our theoretical results: Section 3.1 considers the min-$\ell_2$-norm interpolator and Sections 3.2 and 3.3 consider ridge regression for isotropic and anisotropic covariates, respectively. Sections 3.4.1 and 3.4.2 provide applications of our main results for random effects and spiked covariance models, respectively. Section 4 and presents simulations to complement our theoretical findings. We conclude with a discussion and outlining of future directions in Section 6. Proofs are deferred to the supplementary material.

## 2   PROBLEM SETUP

We consider the fresh data augmentation framework He et al. (2025), but in the context of over-parametrized linear regression. Suppose we observe a dataset $(\mathbf{y}, \mathbf{X})$ from a linear model, i.e.,

$$\mathbf{y} = \mathbf{X}\boldsymbol{\beta} + \boldsymbol{\varepsilon}, \ \text{ with } \mathbf{y}, \boldsymbol{\varepsilon} \in \mathbb{R}^n, \boldsymbol{\beta} \in \mathbb{R}^p, \mathbf{X} \in \mathbb{R}^{n \times p}. \tag{2.1}$$

We compute an initial ridge estimator $\widehat{\boldsymbol{\beta}}_{0,\lambda}$ using $(\mathbf{y}, \mathbf{X})$. For $t \geq 1$ we iteratively generate synthetic response vectors $\widetilde{\mathbf{y}}_{t,\lambda}$ using $\widehat{\boldsymbol{\beta}}_{t-1,\lambda}$, then augment these with fresh real responses $\mathbf{y}_t$. At each step, the next ridge estimator is computed using a mixture of the real and synthetic responses, $\mathbf{y}_t$ and $\widetilde{\mathbf{y}}_{t,\lambda}$ respectively, with a mixing proportion $w \in (0,1)$. The procedure is outlined in Algorithm 1.

---

**Algorithm 1** Iterative ridge with real/synthetic data augmentation

---

1: **Input:** Dataset $(\mathbf{y}, \mathbf{X})$; regularization parameter $\lambda > 0$; mixing proportion $w \in (0,1)$.
2: **Initialize:** $\widehat{\boldsymbol{\beta}}_{0,\lambda} \leftarrow (\mathbf{X}^\top \mathbf{X} + n\lambda \boldsymbol{I})^{-1} \mathbf{X}^\top \mathbf{y}$.
3: **for** $t \geq 1$ **do**
4:     Generate real responses: $\mathbf{y}_t \leftarrow \mathbf{X}\boldsymbol{\beta} + \boldsymbol{\varepsilon}_t$.
5:     Generate synthetic responses: $\widetilde{\mathbf{y}}_{t,\lambda} \leftarrow \mathbf{X}\widehat{\boldsymbol{\beta}}_{t-1,\lambda} + \widetilde{\boldsymbol{\varepsilon}}_t$.
6:     Update estimator:

$$\widehat{\boldsymbol{\beta}}_{t,\lambda} \leftarrow (\mathbf{X}^\top \mathbf{X} + n\lambda \boldsymbol{I})^{-1} \mathbf{X}^\top \big( w\mathbf{y}_t + (1-w)\widetilde{\mathbf{y}}_{t,\lambda} \big).$$

7: **end for**

---

To capture an overparametrized regime, we assume that $p > n$ with both diverging at a comparable rate, i.e. $p/n \to \gamma > 1$. This means we work with a sequence of problem instances $\{\mathbf{y}(n), \mathbf{X}(n), \boldsymbol{\beta}(n), \boldsymbol{\varepsilon}(n)\}_{n \geq 1}$, with $\mathbf{X}(n) \in \mathbb{R}^{n \times p(n)}, \mathbf{y}(n), \boldsymbol{\varepsilon}(n) \in \mathbb{R}^n, \boldsymbol{\beta}(n) \in \mathbb{R}^{p(n)}$, satisfying equation (2.1) and further assume that

$$\lim_{n \to \infty} \|\boldsymbol{\beta}(n)\|^2 = b_\star \in (0, \infty). \tag{2.2}$$

Below we suppress the dependence on $n$ for conciseness. This regime has seen incredible success in modern ML theory in explaining phenomena observed for deep neural networks and other practical algorithms Hastie et al. (2022); Adlam & Pennington (2020); Montanari et al. (2019); Mei & Montanari (2022); Liang & Sur (2022); Cui et al. (2023); Paquette et al. (2024); Emrullah Ildiz et al. (2025); Lu et al. (2025). The regime has also seen enormous utility in high-dimensional statistics, particularly for the development of new theory and methods in challenging contemporary inference problems Bean et al. (2013); El Karoui (2018); Donoho et al. (2009); Bayati & Montanari (2011);

Wang et al. (2017); Sur & Candès (2019); Sur et al. (2019); Fan (2022); Li & Sur (2023); Celentano et al. (2023); Jiang et al. (2025).

In the sequel, we operate under the following assumptions on the covariates and errors that are commonly seen in random matrix theory Bai & Silverstein (2010).

**Assumption 2.1.** *(i) The covariates satisfy* $\mathbf{X} = \mathbf{Z}\mathbf{\Sigma}^{1/2}$*, where* $\mathbf{Z} \in \mathbb{R}^{n \times p}$ *are random matrices whose entries* $Z_{ij}$ *are independent random variables with zero mean and unit variance. We further assume that there exists a constant* $\tau > 0$ *by which the* $\upsilon$*-th moment of each entry is bounded for some* $\upsilon > 4$*, that is,* $\mathbb{E}\left[|Z_{ij}|^{\upsilon}\right] \leq \tau^{-1}$*. We will also assume that* $\mathbf{\Sigma}$ *has bounded eigenvalues* $s_1, \ldots, s_p$*:*

$$\tau \leq s_p \leq \ldots \leq s_1 \leq \tau^{-1}. \tag{2.3}$$

*(ii) The noises* $\boldsymbol{\varepsilon}_t, \widetilde{\boldsymbol{\varepsilon}}_t$ *(defined precisely in Algorithm 1) are assumed to have i.i.d. entries with mean 0, variance* $\sigma^2$*, and bounded moments up to any order. That is, for any* $\phi > 0$*, there exists a constant* $C_\phi$ *such that* $\mathbb{E}\left[|\varepsilon_{t,1}|^\phi\right] \leq C_\phi, \mathbb{E}\left[|\widetilde{\varepsilon}_{t,1}|^\phi\right] \leq C_\phi$*.*

Together with equation (2.2), Assumption 2.1(ii) implies that the signal-to-noise ratio (SNR) $\mathrm{SNR} := b_\star/\sigma^2$ remains finite as $n, p \to \infty$ ensuring that we work under a non-trivial and interesting regime.

## 2.1 RISK

The out-of-sample prediction risk of an estimator $\widehat{\boldsymbol{\beta}}$ (hereafter simply referred to as risk) at a new data point $(y_0, \mathbf{x}_0)$ is defined as

$$R(\widehat{\boldsymbol{\beta}}; \boldsymbol{\beta}) := \mathbb{E}[(\mathbf{x}_0^\top \widehat{\boldsymbol{\beta}} - \mathbf{x}_0^\top \boldsymbol{\beta})^2 | \mathbf{X}] = \mathbb{E}[\|\widehat{\boldsymbol{\beta}} - \boldsymbol{\beta}\|_{\mathbf{\Sigma}}^2 | \mathbf{X}], \tag{2.4}$$

where for a vector $\mathbf{v}$ and matrix $\mathbf{\Sigma}$, we define $\|\mathbf{v}\|_{\mathbf{\Sigma}}^2 = \mathbf{v}^\top \mathbf{\Sigma} \mathbf{v}$. Note that this risk has a $\sigma^2$ difference from the mean-squared prediction error for the new data point, which does not affect the relative performance and is therefore omitted. As defined, the risk involves expectation over both the randomness in the new test point $(y_0, \mathbf{x}_0)$ and that in the noise variables. We define the risk conditional on the feature matrix $\mathbf{X}$, and our risk characterization results are high probability statements over the randomness in the covariates. Despite this dependence on covariates, we use the notation $R(\widehat{\boldsymbol{\beta}}; \boldsymbol{\beta})$ since the context is clear. The risk admits a bias-variance decomposition:

$$R(\widehat{\boldsymbol{\beta}}; \boldsymbol{\beta}) = \underbrace{\|\mathbb{E}(\widehat{\boldsymbol{\beta}}|\mathbf{X}) - \boldsymbol{\beta}\|_{\mathbf{\Sigma}}^2}_{B(\widehat{\boldsymbol{\beta}}; \boldsymbol{\beta})} + \underbrace{\mathrm{Tr}[\mathsf{Cov}(\widehat{\boldsymbol{\beta}}|\mathbf{X})\mathbf{\Sigma}]}_{V(\widehat{\boldsymbol{\beta}}; \boldsymbol{\beta})}. \tag{2.5}$$

We next state our main results, which involve precise characterization of the risk of the estimator $\widehat{\boldsymbol{\beta}}_{t,\lambda}$ from Algorithm 1 and $\widehat{\boldsymbol{\beta}}_t$ defined by equation (3.3).

## 3 MAIN RESULTS

We begin by introducing two measures that feature crucially in the risk of the estimators $\widehat{\boldsymbol{\beta}}_{t,\lambda}$ and $\widehat{\boldsymbol{\beta}}_t$. Let $\boldsymbol{v}_1, \ldots, \boldsymbol{v}_p$ denote the eigenvectors of $\mathbf{\Sigma}$: $\mathbf{\Sigma} = \sum_{k=1}^p s_k \boldsymbol{v}_k \boldsymbol{v}_k^\top$. Define the probability measures:

$$\widehat{H}_p(x) = \frac{1}{p} \sum_{k=1}^p \mathbf{1}_{s_k \leq x}, \qquad \widehat{G}_p(x) = \frac{1}{\|\boldsymbol{\beta}\|_2^2} \sum_{k=1}^p \langle \boldsymbol{v}_k, \boldsymbol{\beta} \rangle^2 \mathbf{1}_{s_k \leq x}. \tag{3.1}$$

Throughout we assume $\widehat{H}_p$ and $\widehat{G}_p$ converge weakly to probability measures $H$ and $G$ respectively. For any $z \in \mathbb{C}/\mathbb{R}^+$, define $m(z)$ to be the solution to

$$m(z)^{-1} + z = \gamma \int \frac{x}{1 + m(z)x} \, dH. \tag{3.2}$$

**Remark 3.1.** *Existence and uniqueness of* $m(z)$ *is well-known (c.f., (Knowles & Yin, 2017, Lemma 2.2)). Further,* $m(z)$ *is the companion Stieltjes transform of the free convolution of* $H$ *and* $MP_\gamma$*, where* $MP_\gamma$ *is the Marchenko-Pastur distribution with parameter* $\gamma$ *Marčenko & Pastur (1967).*

### 3.1 MIN-$\ell_2$-NORM INTERPOLATOR

In this section, we will analyze the behavior of $R(\widehat{\boldsymbol{\beta}}_t; \boldsymbol{\beta})$, where we use

$$\widehat{\boldsymbol{\beta}}_t = (\mathbf{X}^\top \mathbf{X})^\dagger \mathbf{X}^\top (w\mathbf{y}_t + (1-w)\widetilde{\mathbf{y}}_t), \qquad \widetilde{\mathbf{y}}_t = \mathbf{X}\widehat{\boldsymbol{\beta}}_{t-1} + \widetilde{\boldsymbol{\varepsilon}}_t \tag{3.3}$$

in place of $\widehat{\boldsymbol{\beta}}_{t,\lambda}, \widetilde{\mathbf{y}}_{t,\lambda}$ in Algorithm 1. This estimator is a convex combination of the min-$\ell_2$-norm interpolator computed on $(\mathbf{X}, \mathbf{y}_t)$ and $(\mathbf{X}, \widetilde{\mathbf{y}}_t)$. The generalization error of $\widehat{\boldsymbol{\beta}}_t$ is characterized below.

**Theorem 3.1** (Interpolator Risk). *In the setting of Section 2, the risk of $\widehat{\boldsymbol{\beta}}_t$, defined by equation (3.3), satisifes the following. For any $w \in (0,1)$, we have almost surely over the randomness in the covariates,*

$$\lim_{n \to \infty} \lim_{t \to \infty} R(\widehat{\boldsymbol{\beta}}_t; \boldsymbol{\beta}) = \sigma^2 c(w)\mathcal{V} + b_\star \mathcal{B} \tag{3.4}$$

*with $c(w) = (w^2 + (1-w)^2)/w(2-w)$ and*

$$\mathcal{V} = \frac{m'(0)}{m(0)^2} - 1, \qquad \mathcal{B} = \frac{m'(0)}{m(0)^2} \left( \int \frac{x}{(1+m(0)x)^2} dG \right). \tag{3.5}$$

*Moreover, the limiting risk is minimized at $w^\star = \varphi^{-1}$, where $\varphi = (1+\sqrt{5})/2$ is the golden ratio.*

Note Theorem 3.1 applies for any $\boldsymbol{\Sigma}$ obeying our assumptions. For the special case of isotropic features, i.e., $\boldsymbol{\Sigma} = \boldsymbol{I}$, the limiting risk simplifies to

$$\lim_{n \to \infty} \lim_{t \to \infty} R(\widehat{\boldsymbol{\beta}}_t; \boldsymbol{\beta}) = \sigma^2 c(w)\frac{1}{\gamma - 1} + b_\star \left(1 - \frac{1}{\gamma}\right).$$

Proof of Theorem 3.1 is available in Appendix A.

**Effect of number of iteration $t$ and mixing parameter $w$:** In equation (3.4), the first term corresponds to the variance while the second to the bias (recall definitions from equation (2.5)). Note that the quantities $\sigma^2\mathcal{V}$ and $b_\star\mathcal{B}$ coincides asymptotic variance and bias of min-$\ell_2$-norm interpolators in overparametrized regression (Hastie et al., 2022, Theorem 2). In fact, the bias term $B(\widehat{\boldsymbol{\beta}}_t, \boldsymbol{\beta})$ is independent of both the iteration $t$ and mixing proportion $w$. The impact of mixing on the generalization error arises through only the variance, captured by the function $c(w)$, which is minimized at $w^\star = \varphi^{-1} \approx 0.618$. This *golden-ratio weighting* phenomenon was previously observed for a host of models in low-dimensional settings He et al. (2025).

If $w = 0$, then the generalization error $R(\widehat{\boldsymbol{\beta}}_t, \boldsymbol{\beta}) \to \infty$ as $t \to \infty$, even for a finite sample size $n$. This implies that training solely on synthetic data results in model collapse, as seen also by Shumailov et al. (2024); Dohmatob et al. (2024a) for low-dimensional learning problems. Moreover, if the mixing proportion $w > 1/3$, then by equation (A.2), we have the variance $V(\widehat{\boldsymbol{\beta}}_t; \boldsymbol{\beta})$ decreases monotonically with $t$ for any fixed $n$. Since $w^\star = \varphi^{-1} > 1/3$, we observe that the generalization error also decreases monotonically for optimal mixing, thereby preventing model collapse.

**Dynamic mixing:** One might wonder whether the limiting generalization error can be further reduced by selecting the mixing proportion $w_t$ adaptively at each generation to minimize $R(\widehat{\boldsymbol{\beta}}_t; \boldsymbol{\beta})$ for any finite sample size $n$. We show in Section C that the optimal choice $w_t^\star$ in this sequential setup satisfies the recursion

$$w_t^\star = \frac{1 + w_{t-1}^\star}{2 + w_{t-1}^\star}, \quad w_0 = 1.$$

It follows immediately that $w_t^\star$ is decreasing, so if one is free to adjust the mixing proportion at every generation, the optimal strategy places progressively more weight on the synthetic data. Moreover, using $w_t^\star \to w^\star$ (as defined in Theorem 3.1), in the long run, the asymptotic risk is the same whether one uses a fixed $w^\star$ across all generations or adapts $w_t^\star$ dynamically, as proven in Appendix C. Our findings can also be seen empirically in Figure 3 (b), where we plot the risk vs number of iterations for different values of the weight $w$. The difference between $w = \varphi$ and dynamic $w$ is minimal and only visible at $t = 1$. Further, this difference is minimal compared to the difference between $w = 1$ and $w = \varphi^{-1}$. The empirical risks were calculated at $n = 200, p = 400$ with SNR $= 1$.

Next, we study ridge regression. Here, both the variance and bias terms will depend on $\lambda, t, w$.

## 3.2 RIDGE REGRESSION: ISOTROPIC COVARIANCE ($\Sigma = \alpha I$)

In case of isotropic features, i.e., $\Sigma = \alpha I$, the bias and variance of $\widehat{\beta}_{t,\lambda}$ simplifies since $s_k \equiv \alpha$. Hence, by equation (3.1), $G = H = \delta_\alpha$, where $\delta_x$ denote the Dirac probability measure at $x$. This implies $m(z)$, from equation (3.2), simplifies to be the unique solution to

$$m(z)^{-1} + z = \gamma \frac{\alpha}{1 + \alpha m(z)}. \tag{3.6}$$

We use the notations $m_1 := m(-\lambda/w)$, $m_2 := m(-\lambda/(2-w))$.

**Theorem 3.2** (Isotropic risk). *Assume $\Sigma = \alpha I$ and the setting of Section 2. For $0 < w < 1$, $\lambda > 0$, we have almost surely over the randomness in the covariates*

$$\lim_{n\to\infty} \lim_{t\to\infty} R(\widehat{\beta}_{t,\lambda}; \beta) = \sigma^2 c(w) \mathcal{V}_\lambda + b_\star \mathcal{B}_\lambda, \tag{3.7}$$

*where $c(w) = (w^2 + (1-w)^2)/w(2-w)$ and*

$$\mathcal{B}_\lambda = \frac{\alpha/(1+\alpha m_1)^2}{1 - \gamma \alpha^2 m_1^2/(1+\alpha m_1)^2}, \qquad \mathcal{V}_\lambda = \frac{w(2-w)}{2(1-w)} \frac{\gamma}{\lambda} \left( \frac{\alpha}{1+\alpha m_1} - \frac{\alpha}{1+\alpha m_2} \right). \tag{3.8}$$

*Moreover, the limiting risk is a log-convex function of $w$ and has a unique minimizer.*

Theorem 3.2 characterizes the precise asymptotic risk as a function of the regularization parameter $\lambda$ and the mixing proportion $w$. Furthermore, the proof of the result shows that the map $w \mapsto c(w) \mathcal{V}_\lambda$ is log-convex, and the map $w \mapsto \mathcal{B}_\lambda$ is decreasing and log-convex. Hence, we obtain that the limiting risk is log-convex in $w$, thereby admitting a unique minimizer. Further, it can be shown that both $\mathcal{V}_\lambda$ and $\mathcal{B}_\lambda$ are continuous functions of $\lambda$ and $\lim_{\lambda\to 0+} \mathcal{V}_\lambda = \mathcal{V}$ and $\lim_{\lambda\to 0+} \mathcal{B}_\lambda = \mathcal{B}$, with $\mathcal{V}, \mathcal{B}$ defined as in equation (3.5). The proof of Theorem 3.2 shows that the bias at the $t$-th iterate $B(\widehat{\beta}_{t,\lambda}, \beta)$ depends on both $t$ and $w$, unlike the bias of the min-$\ell_2$-norm interpolator. The following result characterizes the behavior of the optimal mixing parameter as a function of $\lambda$.

**Theorem 3.3** (Isotropic Optimal Mixing). *Under the assumptions of Theorem 3.2, let $w^\star(\lambda)$ be the unique global minimizer of the limiting risk as defined in Theorem 3.2. Then $w^\star(\lambda)$ is an increasing function of SNR, a continuous function of $\lambda$ and satisfies (i) $w^\star(\lambda) \in [0.5, 1]$, (ii) $w^\star(\lambda) \to \varphi^{-1} \approx 0.618$ as $\lambda \downarrow 0$ and (iii) $w^\star(\lambda) \to 1$ as $\lambda \uparrow \infty$.*

The optimal mixing parameter minimizing the asymptotic risk is always at least one-half, emphasizing the importance of favoring real-data over synthetic data. In Figure 2 (b), we show empirically that $w^\star$ can be arbitrarily close to $0.5$ for low SNR. Proofs of Theorem 3.2 and Theorem 3.3 are available in Appendix A and Appendix B respectively.

## 3.3 RIDGE REGRESSION: CORRELATED FEATURES

We now state our most general result for anisotropic covariates, which calculates the limiting generalization error of ridge regression for arbitrary measures $\widehat{H}_p, \widehat{G}_p$ (recall equation (3.1)).

**Theorem 3.4** (Ridge risk under correlated covariates). *In the setting of Section 2, suppose $w \in (0,1)$, and $\lambda > 0$. Define $m_1 = m(-\lambda/w)$, $m_2 = m(-\lambda/(2-w))$, where $m(\cdot)$ is the unique solution to equation (3.2). Then almost surely over the randomness in the covariates, equation (3.7) holds with*

$$\mathcal{B}_\lambda = \left( \int \frac{x}{(1+m_1 x)^2} dG \right) \left( 1 - \gamma \int \frac{m_1^2 x^2}{(1+m_1 x)^2} dH \right)^{-1} \tag{3.9}$$

$$\mathcal{V}_\lambda = \frac{w(2-w)}{2(1-w)} \frac{\gamma}{\lambda} \left( \int \frac{x}{1+m_1 x} dH - \int \frac{x}{1+m_2 x} dH \right). \tag{3.10}$$

The proof is deferred to Appendix A. Theorem 3.4 is our most general result. It shows for any $w \in (0,1)$ and any $\lambda > 0$, $R(\widehat{\beta}_{t,\lambda}; \beta)$ does not diverge even when $t$ increases. The result highlights the necessity of mixing real-data with synthetic outputs to mitigate model collapse. In its most general form the risk in equation (3.7) is involved to analyze. In the sequel, we study two popular models where the risk simplifies and the optimal mixing ratio can be studied analytically.

### 3.4 EXAMPLES

In this section, we study two popular classes of examples: (i) the random effects model (Section 3.4.1) and (ii) the spiked covariance model (Section 3.4.2), where structural assumptions on the signal or the population covariance matrix render the limiting risk more tractable.

#### 3.4.1 RANDOM EFFECTS MODEL

Modern applications, ranging from text classification to economics and genomics, are characterized by dense but weak signals spread across many coordinates (Joachims, 1998; Boyle et al., 2017; Yang et al., 2010; Shen & Xiu, 2025). This setting is well captured by a random-effects model, where each feature contributes a small, independent effect, and it provides a simple yet sophisticated framework for rigorously analyzing interesting high-dimensional predictors (Dobriban & Wager, 2018). Adopting a random-effects framework, we assume that each coordinate $\beta_i$ of the signal is drawn i.i.d. with $\mathbb{E}\beta_i = 0$ and $\mathrm{Var}(\beta_i) = b_\star/p > 0$.

**Proposition 3.1** (Ridge risk under Random-Effects Models). *Suppose $\beta_i \overset{i.i.d.}{\sim} (0, b_\star/p)$. Assume the framework of Section 2 and fix $0 < w < 1$, $\lambda > 0$. Let $m(\cdot)$ be the solution to equation (3.2) and define $f(z) = m(-z)^{-1} - z$. Then almost surely over the randomness in the covariates, equation (3.7) holds with*

$$\mathcal{B}_\lambda = \frac{1}{\gamma}\left( f(\lambda/w) - \frac{\lambda}{w} f'(\lambda/w) \right), \quad \mathcal{V}_\lambda = \frac{f\left(\frac{\lambda}{w}\right) - f\left(\frac{\lambda}{2-w}\right)}{\frac{\lambda}{w} - \frac{\lambda}{2-w}}. \tag{3.11}$$

*Moreover, $\mathcal{B}_\lambda$ is decreasing and log-convex and $c(w)\mathcal{V}_\lambda$ is log-convex.*

With this characterization for the risk, we show the optimal mixing ratio satisfies the following.

**Proposition 3.2.** *Under the setup of Theorem 3.1, (i) The generalization error has a unique minimizer $w^\star$, (ii) $w^\star \in [0.5, 1]$, with $w^\star \to 1$ as $\lambda \uparrow \infty$ and $w^\star \to \phi^{-1}$ with $\lambda \downarrow 0$ and (iii) $w^\star$ increases with SNR.*

We provide some context for the random effects model. Our main Theorem 3.4 makes it clear that in presence of general covariance matrices, the variance in the limiting risk (first term in RHS of equation (3.7)) is determined by the spectrum of $\Sigma$, as captured through the limiting spectral distribution $H$. If the signal were deterministic, without additional assumptions, the bias (second term in the RHS of equation (3.7)) would depend on how $\beta$ aligns with the eigenbasis of $\Sigma$. This is captured through the measure $G$. But in a random-effects setting, $\beta$ lies in a generic position relative to $\Sigma$, which simplifies both the limiting bias and variance making equation (3.7) tractable to analyze as a function of $w$. Proofs of Propositions 3.1 and 3.2 are available in Appendix B.

A recent work Dohmatob et al. (2025) studies model collapse in Gaussian random-effects models but they consider a setting where the real and synthetic data are generated from different distributions, subsequently pooling these datasets and studying when model collapse occurs. Crucially, Dohmatob et al. (2025) do not utilize synthetic data generated from a fitted model, unlike in our setting. This leads to fundamental differences in our framework compared to theirs. Additionally, we consider a broad class of random-effects models, without requiring Gaussianity on the signals, and additionally provide guarantees for the optimal mixing ratio.

**Remark 3.2.** *While Propositions 3.1 and 3.2 are stated under the random effects assumption, the conclusions continue to hold for a broader class of parameters $(\beta, \Sigma)$. If the probability measures $\widehat{G}_p$ and $\widehat{H}_p$ from equation (3.1) converge weakly to the same distribution, i.e., $G = H$, we have the same conclusions. In fact, we prove in Lemma D.1 that the random-effects assumption can be seen as a special case of $G = H$. Other natural examples where $G = H$ include the isotropic covariance case, $\Sigma = \alpha I$, or where $\beta$ is drawn uniformly at random from a $p$-dimensional sphere, or $\Sigma$ is drawn from an orthogonally invariant ensemble.*

#### 3.4.2 SPIKED COVARIANCE MODEL

For our second application, we consider a popular class of covariance matrices – the spiked covariance model Birnbaum et al. (2013); Johnstone (2001); Johnstone & Onatski (2020). In the past

two decades, this covariance class has seen exciting applications in population genetics Patterson et al. (2006); Price et al. (2006), finance Knight et al. (2005); Ledoit & Wolf (2022), and signal processing Johnstone & Lu (2009); Wang et al. (2024), among others.

Formally, we assume $\boldsymbol{\Sigma} = \boldsymbol{I} + s\boldsymbol{v}\boldsymbol{v}^\top$ for some $v \in \mathbb{R}^p$, with $\|\boldsymbol{v}\|_2 = 1$, and $s > 0$. The results below extend naturally to spiked models with multiple but finitely many spikes, but for simplicity, we study the problem for the case of a single spike. The limiting risk takes the following form.

**Proposition 3.3.** *In the setting of Section 2, assume that $\boldsymbol{\Sigma} = \boldsymbol{I} + s\boldsymbol{v}\boldsymbol{v}^\top$ and the signal takes the form $\boldsymbol{\beta} = \theta\boldsymbol{v} + \sqrt{1-\theta^2}\boldsymbol{v}^\perp$, while satisfying equation (2.2), with $\boldsymbol{v}^\top\boldsymbol{v}^\perp = 0$ and $\|\boldsymbol{v}^\perp\|_2 = 1$. If $\theta \equiv \theta(n) \to \theta_\star$, then we have almost surely over the randomness of covariates that equation (3.7) holds with $\mathcal{V}_\lambda$ same as in Theorem 3.2 and*

$$\mathcal{B}_\lambda = \left( \theta_\star^2 \frac{1+s}{(1+m_1(1+s))^2} + (1-\theta_\star^2)\frac{1}{(1+m_1)^2} \right) \left( 1 - \gamma \frac{m_1^2}{(1+m_1)^2} \right)^{-1}.$$

*Further, the limiting risk is uniquely minimized at a $w^\star$ satisfying the conclusions of Theorem 3.3.*

Note that, if $\theta_\star \neq 0$, then $G \neq H$. Hence, this spiked matrix case differs fundamentally from the settings discussed in Remark 3.2 and the proof of Proposition 3.3 does not follow directly from the proof of Proposition 3.2. Proof of Proposition 3.3 is available in Appendix B.

# 4 SIMULATIONS

We conduct numerical experiments to complement our theoretical findings. For all plots, we generate $\mathbf{X} = \mathbf{Z}\boldsymbol{\Sigma}^{1/2}$, where $Z_{ij} \sim \mathcal{N}(0,1)$ and vary $\boldsymbol{\Sigma}$ across different plots. We display both the theoretical risk predicted by our formulae (solid lines) and corresponding empirical estimates ($\times$ markers). For the first two plots, the empirical risks are calculated by averaging over 100 runs.

**Risk of min-$\ell_2$-norm Interpolator** In Figure 1, we plot the asymptotic generalization error of the min-$\ell_2$-norm interpolator as a function of mixing weight $w$ for two different classes of covariance matrices: i. $\boldsymbol{\Sigma} = \boldsymbol{I}$ (Panel $(a)$) and ii. $\boldsymbol{\Sigma}_{ij} = 2^{-|i-j|}$ (Panel $(b)$), corresponding to the correlation matrix of an AR(1) process. We vary $\gamma = 1.5, 2, 3$. The choice of the remaining parameters are as follows: sample size $n = 200$, number of features $p = \gamma n$, and number of iterations $t = 5$. To generate $\boldsymbol{\beta}$, we first simulate $\widetilde{\boldsymbol{\beta}}$ with $\widetilde{\beta}_i \overset{\text{i.i.d.}}{\sim} \text{Bern}(0.1)$. Set $\boldsymbol{\beta} = \widetilde{\boldsymbol{\beta}}/\|\widetilde{\boldsymbol{\beta}}\|_2$ which yields $b_\star = 1$.

We observe that the empirical risk matches with its theoretical counterpart even for moderate sample size. Further, the generalization error is always minimized at $w^\star = \varphi^{-1} \approx 0.618$ (dashed vertical line), consistent with Theorem 3.1. Panel $(c)$ shows the risk of the min-$\ell_2$-norm interpolator as a function of iteration $t$. We have computed the risk at optimal mixing weight $w = \varphi^{-1}$ and $\boldsymbol{\Sigma} = \boldsymbol{I}$. We observe that both theoretical and empirical risks stabilize after only a few iterations.

**Optimal weights in ridge regression** In Figure 2, we plot the optimal weight $w^\star$ as a function of $\lambda$. We set $n = 200, p = \gamma n$, and $t = 5$, and vary $\gamma = 1.2, 2, 4$. Figure 2 $(a)$ considers isotropic covariance $\boldsymbol{\Sigma} = \boldsymbol{I}$ with high noise variance $\sigma^2 = 64$. The plot demonstrates that for low SNR, $w^\star$ can be arbitrarily close to $0.5$. Further, $w^\star(\lambda)$ is neither monotone, nor convex as a function of $\lambda$ with $w^\star(\lambda) \to \varphi^{-1}$ as $\lambda \to 0^+$.

Figure 2 $(b)$ corresponds to the spiked covariance model $\boldsymbol{\Sigma} = \boldsymbol{I} + 5\boldsymbol{e}_1\boldsymbol{e}_1^\top$, where $\boldsymbol{e}_1, \ldots, \boldsymbol{e}_p$ denote canonical basis vectors. We set $\beta_1 = 0.5$ and $\boldsymbol{\beta}_{2:p} = \sqrt{1-0.5^2} \times \widetilde{\boldsymbol{\beta}}/\|\widetilde{\boldsymbol{\beta}}\|$ where $\widetilde{\beta}_i \overset{\text{i.i.d.}}{\sim} \text{Bern}(0.25)$. This implies that $\|\boldsymbol{\beta}\|^2 = 1$ and $\theta_\star = 0.5$, where $\theta_\star$ defined as in Proposition 3.3. The plot shows that for large regularization parameter $\lambda$, we have $w^\star = 1$, consistent with Proposition 3.3. Figure 2 $(c)$ plots the risk of $\widehat{\boldsymbol{\beta}}_{t,\lambda}$ with $\sigma^2 = 1, w = \varphi^{-1}$. We take $\boldsymbol{\Sigma} = (1 - \rho/\sqrt{p})I + (\rho/\sqrt{p})\mathbf{1}\mathbf{1}^\top$ with $\rho = 1/2$, and $\beta_i \overset{\text{i.i.d.}}{\sim} \mathcal{N}(0, p^{-1})$. The empirical risk is computed with $\sigma^2 = 1, t = 10, n = 400$. Note that $\boldsymbol{\Sigma}$ does not satisfy bounded eigenvalue condition Assumption 2.1, as its largest eigenvalue $\geq C\sqrt{p}$ for some $C > 0$. Still, the generalization error is accurately predicted by Proposition 3.1 as long as $\boldsymbol{\beta}$ lies in a generic position relative to $\boldsymbol{\Sigma}$. This demonstrates the robustness of our theoretical findings. Figure 3 (a) plots theoretical values of $w^\star$ as a function of $\gamma$ for different values of $\lambda$ (we fixed SNR $= 1/4$). As expected, when $\lambda$ is close to 0, $w^\star \approx \varphi^{-1}$. However, as $\lambda$ increases, the dependence on $\gamma$ becomes more visible. This dependence of the optimal weight on $\gamma$ is important to note, since it illustrates the importance of studying our high-dimensional framework.

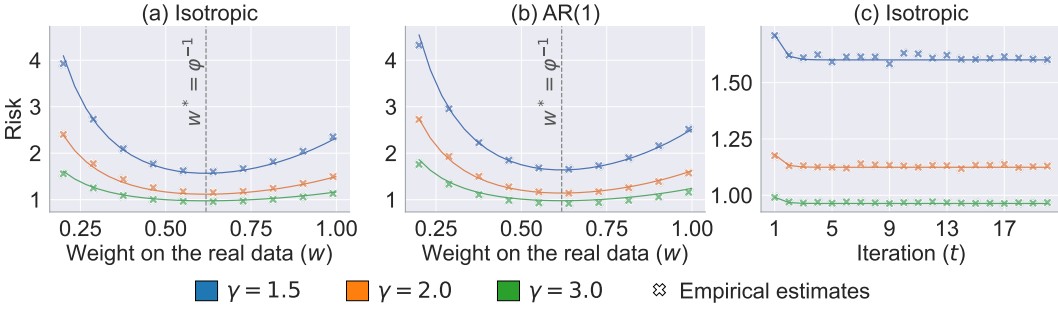

Figure 1: Generalization error of min-$\ell_2$-norm interpolator as a function of weight $w$ (Panel $(a)$ and $(b)$) and iterations $t$ (Panel $(c)$) for different values of $\gamma$. Panel $(a)$ considers isotropic covariance $\boldsymbol{\Sigma} = \boldsymbol{I}$ and panel $(b)$ considers anisotropic $\boldsymbol{\Sigma}$ with $\boldsymbol{\Sigma}_{ij} = 2^{-|i-j|}$, which corresponds to covariance matrix of AR(1) model. In panes $(a)$ and $(b)$, the risk is minimized at $w^\star = 1/\varphi$, as proved by Theorem 3.1. Panel $(c)$ shows that both empirical and theoretical risks stabilize in a few iterations.

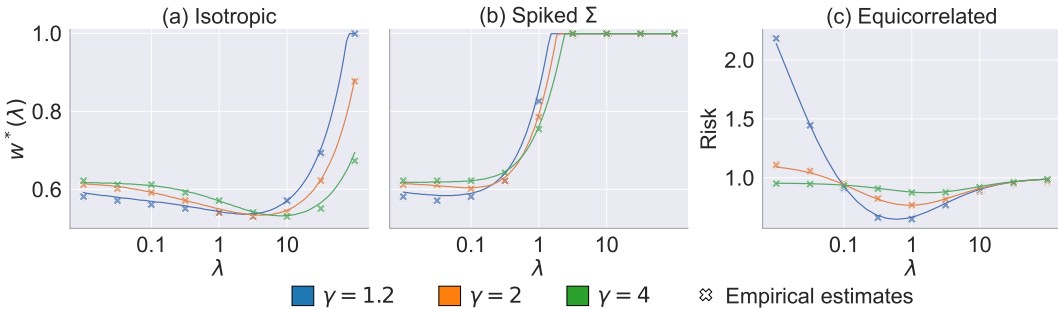

Figure 2: In $(a)$ and $(b)$, we plot the optimal mixing weight $w^\star$ as a function of $\lambda$ for different values of $\gamma$ and two classes of covariance matrices: Panel (a) considers $\boldsymbol{\Sigma} = \boldsymbol{I}$ with high noise $\sigma^2 = 64$, demonstrating $w^\star$ can be close to 0.5 for low SNR. Panel $(b)$ plots $w^\star$ for the spiked covariance matrix showing $w^\star = 1$ for large $\lambda$. Panel $(c)$ plots the generalization error as a function of $\lambda$ for $\boldsymbol{\Sigma}$ equicorrelated matrix. Here, empirical risks align with theoretical predictions given by Proposition 3.1, though $\boldsymbol{\Sigma}$ violates Assumption 2.1, illustrating the robustness of our results.

## 5 BEYOND FRESH DATA AUGMENTATION

In this section, we provide rigorous evidence on when and how model collapse can be prevented beyond the fresh data augmentation framework equation (1.1). In the interest of space, we focus on the min-$\ell_2$-norm interpolator, but our results can be extended for ridge regression.

**No fresh real data** First, we consider the setup where fresh real data is unavailable beyond the original data $(\mathbf{y}, \mathbf{X})$. Therefore, one augments the initial labels $\mathbf{y}$ with synthetic labels as follows

$$\widehat{\boldsymbol{\beta}}_t^{(r)} := (\mathbf{X}^\top \mathbf{X})^\dagger \mathbf{X}^\top \big( w\mathbf{y} + (1-w)\widetilde{\mathbf{y}}_t \big) \text{ where } \widetilde{\mathbf{y}}_t = \mathbf{X}\widehat{\boldsymbol{\beta}}_{t-1}^{(r)} + \widetilde{\boldsymbol{\varepsilon}}_t. \tag{5.1}$$

We assume $\widehat{\boldsymbol{\beta}}_0^{(r)}$ is the min-$\ell_2$-norm interpolator calculated using the original $(\mathbf{X}, \mathbf{y})$.

**Theorem 5.1.** *The risk of $\widehat{\boldsymbol{\beta}}_t^{(r)}$, defined by equation (5.1), satisfies the following. For any $w \in (0, 1)$, we have almost surely over the randomness in the covariates,*

$$\lim_{n\to\infty} \lim_{t\to\infty} R(\widehat{\boldsymbol{\beta}}_t^{(r)}; \boldsymbol{\beta}) = \sigma^2 \frac{1}{w(2-w)} \mathcal{V} + b_\star \mathcal{B} \tag{5.2}$$

*with $\mathcal{V}, \mathcal{B}$ defined as in equation (3.5). The limiting risk is minimized at $w^\star = 1$.*

In Theorem 5.1, $\mathcal{V}, \mathcal{B}$ do not depend on the weight, thus the asymptotic risk is monotonically decreasing in $w$, yielding $w^\star = 1$. Thus, the asymptotic risk is minimized when the weight on the

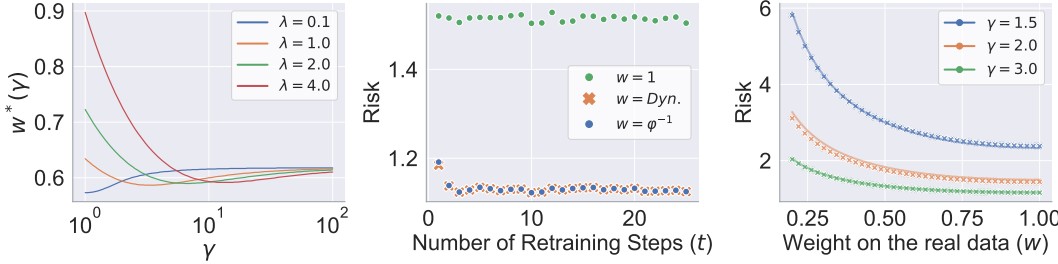

Figure 3: In $(a)$, we plot the optimal $w^\star$ for ridge regression as a function of $\gamma$. In $(b)$, we plot the empirical risk for the min-norm interpolator to demonstrate the dynamic mixing scenario described in Section 3.1. Note that, the risk under dynamic mixing and $w = \varphi^{-1}$ heavily overlap for $t = 2$ onward. In $(c)$, we plot the empirical (the points) & theoretical (the solid lines) risks for the no fresh data augmentation case described by equation (5.1) and Theorem 5.1. Note the risk is optimized at $w^\star = 1$. Throughout $\mathbf{\Sigma} = \mathbf{I}$.

real data is 1, suggesting adding synthetic data following this mechanism fails to improve performance. This can be seen empirically in Figure 3 (c), where we plot the empirical risk as well as the theoretical risk. Here $n = 500$, $\mathrm{SNR} = 1$ and we consider 50 iterations.

**Time-varying covariates** Next, we consider the setup where different covariates are used to train different iterations of the model. Formally, for $t \geq 1$, we have real labels: $\mathbf{y}_t \leftarrow \mathbf{X}_t\boldsymbol{\beta} + \boldsymbol{\varepsilon}_t$ as well as synthetic labels: $\widetilde{\mathbf{y}}_t \leftarrow \mathbf{X}_t\widehat{\boldsymbol{\beta}}_{t-1} + \widetilde{\boldsymbol{\varepsilon}}_t$. The estimator for $t$-th iteration is defined as

$$\widehat{\boldsymbol{\beta}}_t^{(v)} := (\mathbf{X}_t^\top\mathbf{X}_t)^\dagger\mathbf{X}_t^\top\left(w\mathbf{y}_t + (1 - w)\widetilde{\mathbf{y}}_t\right). \tag{5.3}$$

We provide the precise generalization error of $\widehat{\boldsymbol{\beta}}_t^{(v)}$ in Theorem E.1 (for simplicity we assume $\mathbf{\Sigma} = \mathbf{I}$, see also Remark E.1 regarding limit orders) and characterize the unique optimal weight at which the risk is minimized. Unlike Theorem 3.1, both the bias and variance depend on $w$.

**Time-varying covariates with fresh labels in a fraction** Finally, we consider a setting where at each iteration, fresh labels are available only for a fraction of covariates. Since all covariates do not both see real and synthetic labels, instead of our mixing framework from equation (1.1), we consider pooling the datasets with real and synthetic labels. Formally, for $t \geq 1$, we observe real responses: $\mathbf{y}_t \leftarrow \mathbf{X}_t\boldsymbol{\beta} + \boldsymbol{\varepsilon}_t$ and synthetic responses: $\widetilde{\mathbf{y}}_t \leftarrow \widetilde{\mathbf{X}}_t\widehat{\boldsymbol{\beta}}_{t-1} + \widetilde{\boldsymbol{\varepsilon}}_t$. Define

$$\widehat{\boldsymbol{\beta}}_t^{(a)} \leftarrow (\mathbf{X}_t^\top\mathbf{X}_t + \widetilde{\mathbf{X}}_t^\top\widetilde{\mathbf{X}}_t)^\dagger\left(\mathbf{X}_t^\top\mathbf{y}_t + \widetilde{\mathbf{X}}_t^\top\widetilde{\mathbf{y}}_t\right).$$

In Appendix E.3 (specifically Theorem E.2), we show that there exist ranges of values of $\gamma$ where preventing model collapse is not possible in this setting.

## 6 DISCUSSION

We provide a rigorous analysis of model collapse under overparametrization for linear models. As an overarching theme, we demonstrate how mixing real-data with synthetic outputs mitigates model collapse, and identify optimal mixing ratios that minimize prediction error in this context. As a promising next direction, understanding how model collapse affects interpolators in other $\ell_p$ geometries would be crucial. Such interpolators arise as implicit regularized limits of popular algorithms Gunasekar et al. (2018b) and typically require techniques beyond random matrix theory—a critical tool employed in our analysis. Additionally, approaches that extend linear arguments to non-linear high-dimensional problems (c.f., Sur & Candès (2019); Hu & Lu (2022)) should enable our qualitative conclusions to generalize to structured non-linear models. This includes generalized linear models, single-index models, and even non-parametric models through parametric-to-non-parametric equivalence techniques introduced in Lahiry & Sur (2023). Finally, extending our results to more complex architectures remains an important challenge. One promising approach involves studying multi-index models and their sequential variants, which capture classes of neural networks and transformer architectures Troiani et al. (2024; 2025); Cui (2025). Investigating model collapse and mitigation strategies in these contexts presents an exciting avenue for future research.

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

## Supplementary Material

## A PROOF OF THEOREM 3.1, THEOREM 3.2, AND THEOREM 3.4

### A.1 NOTATIONS

Our objective is to derive the asymptotic generalization error of $\widehat{\boldsymbol{\beta}}_{t,\lambda}$ given by Algorithm 1 and $\widehat{\boldsymbol{\beta}}_t$ defined by equation (3.3). We define the scaled sample covariance matrix and its resolvent as

$$\mathbf{M} := \mathbf{X}^\top \mathbf{X}, \qquad \mathbf{A}_\lambda := (\mathbf{M} + n\lambda \boldsymbol{I})^{-1}.$$

Note that $\mathbf{A}_\lambda$ and $\mathbf{M}$ are simultaneously diagonalizable and thus commute with each other. If $\lambda = 0$, we use the notation $\mathbf{A}_0 = (\mathbf{X}^\top \mathbf{X})^\dagger$. The mixing proportion of the real-data is denoted by $w$ and we define $\widetilde{w} := 1 - w$. For two sequence of random variables $u_n, v_n$, we use the notation $u_n \sim v_n$ if $u_n/v_n \xrightarrow{P} 1$ as $n \to \infty$.

### A.2 VARIANCE

We use the notations of Subsection A.1 throughout the proof. First, we consider the case $\lambda > 0$.

**Lemma A.1** (Ridge $\widehat{\boldsymbol{\beta}}_{t,\lambda}$ Covariance). *For $\lambda > 0$, the covariance matrix $\mathsf{Cov}[\widehat{\boldsymbol{\beta}}_{t,\lambda}|\mathbf{X}]$ is given by*

$$\sigma^2 \left[ (w^2 + \widetilde{w}^2) \sum_{k=1}^t \widetilde{w}^{2(k-1)} \mathbf{M}^{2k-1} \mathbf{A}_\lambda^{2k} + \widetilde{w}^{2t} \mathbf{M}^{2t+1} \mathbf{A}_\lambda^{2t+2} \right] \tag{A.1}$$

*Proof of Lemma A.1.* We will prove equation (A.1) by induction. For the base case $(t = 0)$, we have $\widehat{\boldsymbol{\beta}}_{0,\lambda} = \mathbf{A}_\lambda \mathbf{X}^\top \mathbf{y}$. Hence,

$$\mathsf{Cov}[\widehat{\boldsymbol{\beta}}_{0,\lambda}|\mathbf{X}] = \mathbf{A}_\lambda \mathbf{X}^\top \mathsf{Cov}[\mathbf{y}|\mathbf{X}] \mathbf{X} \mathbf{A}_\lambda$$
$$= \mathbf{A}_\lambda \mathbf{X}^\top (\sigma^2 \boldsymbol{I}) \mathbf{X} \mathbf{A}_\lambda = \sigma^2 \mathbf{A}_\lambda^2 \mathbf{M}$$

This proves the base case. Now let's assume equation (A.1) holds for some $t$. Then, for iteration $(t + 1)$, we have

$$\mathsf{Cov}[\widehat{\boldsymbol{\beta}}_{t+1,\lambda}|\mathbf{X}] = \mathbf{A}_\lambda \mathbf{X}^\top \mathsf{Cov}[w\mathbf{y}_t + \widetilde{w}\widetilde{\mathbf{y}}_t|\mathbf{X}] \mathbf{X} \mathbf{A}_\lambda$$
$$= \mathbf{A}_\lambda \mathbf{X}^\top (w^2 \sigma^2 I + \widetilde{w}^2 \mathbf{X} \mathsf{Cov}[\widehat{\boldsymbol{\beta}}_{t,\lambda}|\mathbf{X}] \mathbf{X}^\top + \sigma^2 \widetilde{w}^2 I) \mathbf{X} \mathbf{A}_\lambda$$
$$= \widetilde{w}^2 \mathbf{A}_\lambda \mathbf{M} \mathsf{Cov}[\widehat{\boldsymbol{\beta}}_{t,\lambda}|\mathbf{X}] \mathbf{M} \mathbf{A}_\lambda + \sigma^2 (w^2 + \widetilde{w}^2) \mathbf{M} \mathbf{A}_\lambda^2$$
$$= \widetilde{w}^2 \mathbf{A}_\lambda \mathbf{M} \left( \sigma^2 \left[ (w^2 + \widetilde{w}^2) \sum_{k=1}^t \widetilde{w}^{2(k-1)} \mathbf{M}^{2k-1} \mathbf{A}_\lambda^{2k} + \widetilde{w}^{2t} \mathbf{M}^{2t+1} \mathbf{A}_\lambda^{2t+2} \right] \right) \mathbf{M} \mathbf{A}_\lambda$$
$$\quad + \sigma^2 (w^2 + \widetilde{w}^2) \mathbf{M} \mathbf{A}_\lambda^2$$
$$= \sigma^2 \left[ (w^2 + \widetilde{w}^2) \sum_{k=1}^t \widetilde{w}^{2k} \mathbf{M}^{2k+1} \mathbf{A}_\lambda^{2k+2} + \widetilde{w}^{2t+2} \mathbf{M}^{2t+3} \mathbf{A}_\lambda^{2t+4} \right]$$
$$\quad + \sigma^2 (w^2 + \widetilde{w}^2) \mathbf{M} \mathbf{A}_\lambda^2$$
$$= \sigma^2 \left[ (w^2 + \widetilde{w}^2) \sum_{k=1}^{t+1} \widetilde{w}^{2(k-1)} \mathbf{M}^{2k-1} \mathbf{A}_\lambda^{2k} + \widetilde{w}^{2t+2} \mathbf{M}^{2t+3} \mathbf{A}_\lambda^{2t+4} \right]$$

This completes the proof of equation (A.1). □

**Corollary A.1.** *Using equation (2.5), $V(\widehat{\boldsymbol{\beta}}_{t,\lambda}; \boldsymbol{\beta})$ is given by*

$$V(\boldsymbol{\beta}_{t,\lambda}; \boldsymbol{\beta}) = \sigma^2 \left[ (w^2 + \widetilde{w}^2) \sum_{k=1}^t \widetilde{w}^{2(k-1)} \operatorname{Tr}[\mathbf{B}_\lambda^k] + \widetilde{w}^{2t} \operatorname{Tr}[\mathbf{B}_\lambda^{t+1}] \right] \tag{A.2}$$

*where $\mathbf{B}_\lambda^k = \mathbf{M}^{2k-1} \mathbf{A}_\lambda^{2k} \boldsymbol{\Sigma}$.*

Next, note that $\sigma_{\max}(\mathbf{M}\mathbf{A}_\lambda) < 1$ and therefore $\mathsf{Cov}[\widehat{\boldsymbol{\beta}}_{t,\lambda}|\mathbf{X}]$ converges in matrix norm to the following limit:

$$
\begin{aligned}
\lim_{t\to\infty} \mathsf{Cov}[\widehat{\boldsymbol{\beta}}_{t,\lambda}|\mathbf{X}] &= \sigma^2(w^2+(1-w)^2)\sum_{k=0}^{\infty}(1-w)^{2k}(\mathbf{X}^\top\mathbf{X})^{2k+1}\mathbf{A}_\lambda^{2k+2} \\
&= \sigma^2(w^2+(1-w)^2)\mathbf{X}^\top\mathbf{X}\mathbf{A}_\lambda^2(I-(1-w)^2(\mathbf{X}^\top\mathbf{X})^2A_\lambda^2)^{-1} \\
&= \sigma^2(w^2+\widetilde{w}^2)\mathbf{X}^\top\mathbf{X}(n\lambda+w\mathbf{X}^\top\mathbf{X})^{-1}(n\lambda+(2-w)\mathbf{X}^\top\mathbf{X})^{-1} \\
&= \frac{\sigma^2(w^2+(1-w)^2)}{2(1-w)}\left[(w\mathbf{X}^\top\mathbf{X}+n\lambda\boldsymbol{I})^{-1}-((2-w)\mathbf{X}^\top\mathbf{X}+n\lambda\boldsymbol{I})^{-1}\right]
\end{aligned}
$$

$$
\implies \lim_{t\to\infty} V(\widehat{\boldsymbol{\beta}}_{t,\lambda};\boldsymbol{\beta}) = \frac{\sigma^2(w^2+(1-w)^2)}{2(1-w)}\left(\frac{1}{w}\mathrm{Tr}[\mathbf{A}_{\lambda/w}\boldsymbol{\Sigma}]-\frac{1}{2-w}\mathrm{Tr}[\mathbf{A}_{\lambda/(2-w)}\boldsymbol{\Sigma}]\right)
$$

where the second to last equality follows using the following equality

$$
\frac{x}{(n\lambda+wx)(n\lambda+(2-w)x)} = \frac{1}{2(1-w)}\left(\frac{1}{n\lambda+wx}-\frac{1}{n\lambda+(2-w)x}\right),
$$

combined with diagonalization of $\mathbf{X}^\top\mathbf{X}$. Thus, to compute the limiting variance, it is enough to calculate

$$
\lim_{n\to\infty} \mathrm{Tr}[\mathbf{A}_{-z}\boldsymbol{\Sigma}] = \lim_{n\to\infty}\frac{1}{n}\mathrm{Tr}\left[(n^{-1}\mathbf{X}^\top\mathbf{X}-z\boldsymbol{I}_p)^{-1}\boldsymbol{\Sigma}\right],
$$

for $z\in\mathbb{C}/\mathbb{R}^+$. Using averaged local law Knowles & Yin (2017), we obtain

$$
\frac{1}{n}\mathrm{Tr}\left[(n^{-1}\mathbf{X}^\top\mathbf{X}-z\boldsymbol{I}_p)^{-1}\boldsymbol{\Sigma}\right] \xrightarrow{\text{a.s.}} \frac{-1}{z}\gamma\int\frac{x}{mx+1}dH(x),
$$

where $m(z)$ is as defined in equation (3.2). Combining the above display with Corollary A.1, we obtain that

$$
\lim_{n\to\infty}\lim_{t\to\infty} V(\widehat{\boldsymbol{\beta}}_t;\boldsymbol{\beta}) = \sigma^2\frac{\gamma}{\lambda}\frac{(w^2+(1-w)^2)}{2(1-w)}\left[\int\frac{x}{1+m_1x}dH-\int\frac{x}{1+m_2x}dH\right], \tag{A.3}
$$

where $m_1$ and $m_2$ are as defined in equation (3.10). This completes the proof of the expression of variance for $\lambda > 0$.

Next, we turn our attention to the case $\lambda = 0$. Here, we have the following expression of covariance.

**Lemma A.2.** *For $t \geq 1$, the covariance matrix $\mathsf{Cov}[\widehat{\boldsymbol{\beta}}_t|\mathbf{X}]$ is given by*

$$
\mathsf{Cov}[\widehat{\boldsymbol{\beta}}_t|\mathbf{X}] = \sigma^2(\mathbf{X}^\top\mathbf{X})^\dagger\left[(w^2+(1-w)^2)\sum_{k=1}^{t}(1-w)^{2(k-1)}+(1-w)^{2t}\right], \tag{A.4}
$$

*where $\widehat{\boldsymbol{\beta}}_t$ is given by equation (3.3).*

The proof follows the same steps as proof of Lemma A.1 once we note that $(\mathbf{X}^\top\mathbf{X})^{2k-1}((\mathbf{X}^\top\mathbf{X})^\dagger)^{2k} = (\mathbf{X}^\top\mathbf{X})^\dagger$ for any $k \geq 1$.

As a consequence, we have

$$
\lim_{t\to\infty} V(\widehat{\boldsymbol{\beta}}_t;\boldsymbol{\beta}) = \sigma^2\frac{w^2+(1-w)^2}{w(2-w)}\mathrm{Tr}[(\mathbf{X}^\top\mathbf{X})^\dagger\boldsymbol{\Sigma}] \tag{A.5}
$$

The limit of the above quantity as $n \to \infty$ is given by (Hastie et al., 2022, Theorem 2) and is equal to the variance given by equation (3.5).

## A.3   BIAS

We use the notation of Subsection A.1 throughout this subsection.

**Lemma A.3** (Ridge $\widehat{\boldsymbol{\beta}}_{t,\lambda}$ Expectation). *For $\lambda > 0$, we have*

$$\mathbb{E}[\widehat{\boldsymbol{\beta}}_{t,\lambda}|\mathbf{X}] = (\widetilde{w}\mathbf{A}_\lambda\mathbf{M})^t\mathbf{A}_\lambda\mathbf{M}\boldsymbol{\beta} + w\sum_{i=0}^{t-1}(\widetilde{w}\mathbf{A}_\lambda\mathbf{M})^i\mathbf{A}_\lambda\mathbf{M}\boldsymbol{\beta} \tag{A.6}$$

*Proof.* We prove equation (A.6) by induction. Let's start with the base case ($t = 0$).

$$\begin{aligned}
\mathbb{E}[\widehat{\boldsymbol{\beta}}_{0,\lambda}|\mathbf{X}] &= (\mathbf{X}^\top\mathbf{X} + n\lambda)^{-1}\mathbf{X}^\top\mathbb{E}[\mathbf{y}|\mathbf{X}] \\
&= \mathbf{A}_\lambda\mathbf{M}\boldsymbol{\beta}
\end{aligned}$$

This proves the base case. Now assume equation (A.6) holds for $t = k$. Let's prove it for $t = k + 1$.

$$\begin{aligned}
\mathbb{E}[\widehat{\boldsymbol{\beta}}_{k+1,\lambda}|\mathbf{X}] &= \mathbf{A}_\lambda\mathbf{M}(w\boldsymbol{\beta} + (1-w)\mathbb{E}[\widehat{\boldsymbol{\beta}}_{k,\lambda}|\mathbf{X}]) \\
&= w\mathbf{A}_\lambda\mathbf{M}\boldsymbol{\beta} + \widetilde{w}\mathbf{A}_\lambda\mathbf{M}\mathbb{E}[\widehat{\boldsymbol{\beta}}_{k,\lambda}|\mathbf{X}]) \\
&= w\mathbf{A}_\lambda\mathbf{M}\boldsymbol{\beta} + \widetilde{w}\mathbf{A}_\lambda\mathbf{M}\left[(\widetilde{w}\mathbf{A}_\lambda\mathbf{M})^k\mathbf{A}_\lambda\mathbf{M}\boldsymbol{\beta} + w\sum_{i=0}^{k-1}(\widetilde{w}\mathbf{A}_\lambda\mathbf{M})^i\mathbf{A}_\lambda\mathbf{M}\boldsymbol{\beta}\right] \\
&= (\widetilde{w}\mathbf{A}_\lambda\mathbf{M})^{k+1}\mathbf{A}_\lambda\mathbf{M}\boldsymbol{\beta} + w\sum_{i=0}^{k}(\widetilde{w}\mathbf{A}_\lambda\mathbf{M})^i\mathbf{A}_\lambda\mathbf{M}\boldsymbol{\beta}.
\end{aligned}$$

This completes the proof of Lemma A.3. $\qquad\square$

**Corollary A.2.** *Bias for the Ridge Estimator is given by*

$$B(\widehat{\boldsymbol{\beta}}_{t,\lambda};\boldsymbol{\beta}) = \left(\frac{n\lambda}{w}\right)^2\|\mathbf{A}_{\lambda/w}(\boldsymbol{I} - (\widetilde{w}\mathbf{A}_\lambda\mathbf{M})^{t+1})\boldsymbol{\beta}\|_{\boldsymbol{\Sigma}}^2. \tag{A.7}$$

$$\lim_{t\to\infty} B(\widehat{\boldsymbol{\beta}}_{t,\lambda};\boldsymbol{\beta}) = \left(\frac{n\lambda}{w}\right)^2\|\mathbf{A}_{\lambda/w}\boldsymbol{\beta}\|_{\boldsymbol{\Sigma}}^2 = \left(\frac{n\lambda}{w}\right)^2\text{Tr}[\boldsymbol{\beta}\boldsymbol{\beta}^\top\mathbf{A}_{\lambda/w}\boldsymbol{\Sigma}\mathbf{A}_{\lambda/w}]. \tag{A.8}$$

*Proof.* From equation (A.6), we have

$$\begin{aligned}
\boldsymbol{\beta} - \mathbb{E}[\widehat{\boldsymbol{\beta}}_{t,\lambda}|\mathbf{X}] &= \boldsymbol{\beta} - (\widetilde{w}\mathbf{A}_\lambda\mathbf{M})^t\mathbf{A}_\lambda\mathbf{M}\boldsymbol{\beta} \\
&\quad - w(I - \widetilde{w}\mathbf{A}_\lambda\mathbf{M})^{-1}(I - (\widetilde{w}\mathbf{A}_\lambda\mathbf{M})^t)\mathbf{A}_\lambda\mathbf{M}\boldsymbol{\beta}. \tag{A.9}
\end{aligned}$$

Diagonalizing $\mathbf{A}_\lambda\mathbf{M}$ and simplifying using the following identity

$$1 - \widetilde{w}^t x^{t+1} - w\frac{1 - \widetilde{w}^t x^t}{1 - \widetilde{w}x}x = \frac{(1-x)(1 - \widetilde{w}^{1+t}x^{1+t})}{1 - \widetilde{w}x},$$

tells us that

$$\boldsymbol{\beta} - \mathbb{E}[\widehat{\boldsymbol{\beta}}_t|\mathbf{X}] = (\boldsymbol{I} - \mathbf{A}_\lambda\mathbf{M})(\boldsymbol{I} - \widetilde{w}\mathbf{A}_\lambda\mathbf{M})^{-1}(\boldsymbol{I} - (\widetilde{w}A_\lambda\mathbf{M})^{1+t})\boldsymbol{\beta}$$

Next, we diagonalize $M$ use the following identity on product of first two matrices.

$$\left(1 - \frac{x}{x + n\lambda}\right)\left(1 - \frac{\widetilde{w}x}{x + n\lambda}\right)^{-1} = \frac{n\lambda}{wx + n\lambda} = \frac{n\lambda}{w}\left(x + n\frac{\lambda}{w}\right)^{-1}$$

to conclude $(\boldsymbol{I} - \mathbf{A}_\lambda\mathbf{M})(\boldsymbol{I} - \widetilde{w}\mathbf{A}_\lambda\mathbf{M})^{-1} = \frac{n\lambda}{w}\mathbf{A}_{\lambda/w}$. This completes the proof of equation (A.7). equation (A.8) follows since $\sigma_{\max}(\widetilde{w}\mathbf{A}_\lambda\mathbf{M}) < \widetilde{w} < 1$. $\qquad\square$

**Lemma A.4.** *Suppose Assumption 2.1 holds. For any deterministic sequence of symmetric matrices $C \in \mathbb{R}^{p\times p}$ with bounded operator norm and $z \in \mathbb{C} \setminus \mathbb{R}^+$, we have*

$$z^2\,\text{Tr}[\boldsymbol{C}\boldsymbol{Q}_n(z)\boldsymbol{\Sigma}\boldsymbol{Q}_n(z)] \sim \text{Tr}[\boldsymbol{C}(\boldsymbol{I} + m(z)\boldsymbol{\Sigma})^{-2}\boldsymbol{\Sigma}]\frac{1}{1 - \frac{1}{n}df_2(1/m(z))} \tag{A.10}$$

*where $df_2(\kappa) = \text{Tr}[\boldsymbol{\Sigma}^2(\kappa\boldsymbol{I} + \boldsymbol{\Sigma})^{-2}]$ and $\boldsymbol{Q}_n(z) = (n^{-1}\mathbf{M} - z\boldsymbol{I})^{-1}$.*

*Proof.* By plugging in $A = C$ and $B = \boldsymbol{\Sigma}$ in (Bach, 2024, Eq (3.9)) and noticing that $(\boldsymbol{I} + m(z)\boldsymbol{\Sigma})$ and $\boldsymbol{\Sigma}$ are simultaneously diagonalizable (and therefore commute), we get

$$z^2 \operatorname{Tr}[CQ_n(z)\boldsymbol{\Sigma}Q_n(z)] \sim \operatorname{Tr}[C(\boldsymbol{I} + m(z)\boldsymbol{\Sigma})^{-2}\boldsymbol{\Sigma}]$$

$$+ \operatorname{Tr}[C(\boldsymbol{I} + m(z)\boldsymbol{\Sigma})^{-2}\boldsymbol{\Sigma}] \cdot \frac{\operatorname{Tr}[\boldsymbol{\Sigma}^2 \left(m(z)^{-1}\boldsymbol{\Sigma} + \boldsymbol{I}\right)^{-2}]}{n - df_2(m(z)^{-1})}$$

$$\sim \operatorname{Tr}[C(\boldsymbol{I} + m(z)\boldsymbol{\Sigma})^{-2}\boldsymbol{\Sigma}] \cdot \left(1 + \frac{df_2(m(z)^{-1})}{n - df_2(m(z)^{-1})}\right)$$

$$\sim \operatorname{Tr}[C(\boldsymbol{I} + m(z)\boldsymbol{\Sigma})^{-2}\boldsymbol{\Sigma}] \cdot \frac{1}{1 - n^{-1}df_2(m(z)^{-1})}$$

This completes the proof of equation (A.10). $\square$

We now show that Equation (3.9) holds. Start by noticing that $n\mathbf{A}_z = Q_n(-z)$. Then, equation (A.7) combined with Lemma A.4 tells us that

$$\lim_{t \to \infty} B(\widehat{\boldsymbol{\beta}}_{t,\lambda}; \boldsymbol{\beta}) = \left(\frac{n\lambda}{w}\right)^2 \operatorname{Tr}[\boldsymbol{\beta}\boldsymbol{\beta}^\top \mathbf{A}_{\lambda/w}\boldsymbol{\Sigma}\mathbf{A}_{\lambda/w}]$$

$$\sim \operatorname{Tr}[\boldsymbol{\beta}\boldsymbol{\beta}^\top (I + m_1\boldsymbol{\Sigma})^{-2}\boldsymbol{\Sigma}] \cdot \frac{1}{1 - n^{-1}df_2(m_1^{-1})}$$

where $m_1 = m(-\lambda/w)$. All that is left to get equation (3.9) is realizing that

$$\lim_{n \to \infty} \operatorname{Tr}[\boldsymbol{\beta}\boldsymbol{\beta}^\top (\boldsymbol{I} + m_1\boldsymbol{\Sigma})^{-2}\boldsymbol{\Sigma}] = b_\star \int \frac{x}{(1 + m_1 x)^2} dG,$$

$$\text{and } \lim_{n \to \infty} n^{-1}df_2(m_1^{-1}) = \gamma \int \frac{m_1^2 x^2}{(1 + m_1 x)^2} dH. \tag{A.11}$$

This completes the proof of limiting value of bias.

PROOF OF THEOREM 3.4

The asymptotic variance of $\widehat{\boldsymbol{\beta}}_{t,\lambda}$ is given by equation (A.3) for general $\boldsymbol{\Sigma}$. The asymptotic bias is given by equation (A.11). This completes the proof of Theorem 3.4.

PROOF OF THEOREM 3.2

Plugging $H = G = \delta_\alpha$ in equation (3.10) and equation (3.9) yields the asymptotic variance and bias for $\widehat{\boldsymbol{\beta}}_{t,\lambda}$ when $\boldsymbol{\Sigma} = \boldsymbol{I}$. The proof of log-convexity of risk follows from the proof of Theorem 3.3

PROOF OF THEOREM 3.1

The asymptotic variance of $\widehat{\boldsymbol{\beta}}_t$ is obtained by equation (A.5). To obtain the bias, recall that $\widehat{\boldsymbol{\beta}}_t = (\mathbf{X}^\top\mathbf{X})^\dagger\mathbf{X}^\top(w\mathbf{y}_t + (1-w)\widetilde{\mathbf{y}}_t)$. Since $\widehat{\boldsymbol{\beta}}_0 = (\mathbf{X}^\top\mathbf{X})^\dagger\mathbf{X}^\top\mathbf{y}$, we have $\mathbb{E}(\widehat{\boldsymbol{\beta}}_0|\mathbf{X}) = (\mathbf{X}^\top\mathbf{X})^\dagger\mathbf{X}^\top\mathbf{X}\boldsymbol{\beta}$. Now, we want to prove by induction that

$$\mathbb{E}(\widehat{\boldsymbol{\beta}}_t|\mathbf{X}) = (\mathbf{X}^\top\mathbf{X})^\dagger\mathbf{X}^\top\mathbf{X}\boldsymbol{\beta} \tag{A.12}$$

for all $t$. To this end note that,

$$\mathbb{E}(\widehat{\boldsymbol{\beta}}_t|\mathbf{X}) = (\mathbf{X}^\top\mathbf{X})^\dagger\mathbf{X}^\top\left(w\mathbb{E}(\mathbf{y}_t|\mathbf{X}) + (1-w)\mathbb{E}(\widetilde{\mathbf{y}}_t|\mathbf{X})\right). \tag{A.13}$$

Since $\mathbf{y}_t = \mathbf{X}\boldsymbol{\beta} + \boldsymbol{\varepsilon}_t$, we have $\mathbb{E}(\mathbf{y}_t|\mathbf{X}) = \mathbf{X}\boldsymbol{\beta}$. For the synthetic data, $\widetilde{\mathbf{y}}_t = \mathbf{X}\widehat{\boldsymbol{\beta}}_{t-1} + \widehat{\boldsymbol{\varepsilon}}_t$. Therefore,

$$\mathbb{E}(\widetilde{\mathbf{y}}_t|\mathbf{X}) = \mathbf{X}\mathbb{E}(\widehat{\boldsymbol{\beta}}_{t-1}|\mathbf{X}) = \mathbf{X}(\mathbf{X}^\top\mathbf{X})^\dagger\mathbf{X}^\top\mathbf{X}\boldsymbol{\beta}$$

by induction hypothesis. Using equation (A.13) we have

$$\mathbb{E}(\widehat{\boldsymbol{\beta}}_t|\mathbf{X}) = (\mathbf{X}^\top\mathbf{X})^\dagger\mathbf{X}^\top\left(w\mathbf{X}\boldsymbol{\beta} + (1-w)\mathbf{X}(\mathbf{X}^\top\mathbf{X})^\dagger\mathbf{X}^\top\mathbf{X}\boldsymbol{\beta}\right) = (\mathbf{X}^\top\mathbf{X})^\dagger\mathbf{X}^\top\mathbf{X}\boldsymbol{\beta}.$$

This proves equation (A.12) by induction. This in turn implies using equation (2.5) that for any $n, t$,

$$B(\widehat{\boldsymbol{\beta}}_t; \boldsymbol{\beta}) = \boldsymbol{\beta}^\top P_{\mathbf{X}} \boldsymbol{\Sigma} P_{\mathbf{X}} \boldsymbol{\beta},$$

where $P_{\mathbf{X}} = \boldsymbol{I} - (\mathbf{X}^\top \mathbf{X})^\dagger (\mathbf{X}^\top \mathbf{X})$. Using (Hastie et al., 2022, Theorem 2), we obtain the analytic value of the bias.

Finally, note that the asymptotic risk depends on $w$ only via $c(w)$. Since $c(w)$ is minimized at $1/\varphi$, this completes the proof of Theorem 3.1.

## B  PROOFS OF REMAINING RESULTS

In this Section, we will show that the expression of generalization error $R(\widehat{\boldsymbol{\beta}}_{t,\lambda})$ can be simplified further if the probability measures $\widehat{G}_p$ and $\widehat{H}_p$ defined by equation (3.1) weakly converges to the same probability distribution, i.e., $G = H$. In this special case, the generalization error has unique minima w.r.to $w$.

Recall the definitions of $\mathcal{V}_\lambda$ and $\mathcal{B}_\lambda$ given by equation (3.10) and equation (3.9) respectively. We will first rewriting $\mathcal{V}_\lambda$ and $\mathcal{B}_\lambda$ in a different form. To this end, differentiate both sides of equation (3.2) w.r.to $z$ to obtain

$$-\frac{m'(z)}{m^2(z)} + 1 = \gamma \int \frac{-x^2 m'(z)}{(1 + xm(z))^2} dH. \tag{B.1}$$

The above equality will be helpful in writing the integrals concisely. Also define $f(z) = m(-z)^{-1} - z$. We know that $m(z)$ is a Stieltjes transform of a non-negative random variable by Remark 3.1. We will need the following technical Lemma whose proof we defer.

**Lemma B.1.** *There exists some measure $\mu$ on $\mathbb{R}^+$ with $|f(1)| < \infty$ such that*

$$f(z) = a + \int \frac{z}{z + t} \mu(dt). \tag{B.2}$$

Invoking Lemma B.1, equation (3.10) tells us that

$$\frac{2(1 - w)}{w(2 - w)} \mathcal{V}_\lambda = \frac{\gamma}{\lambda} \left( \int \frac{x}{1 + m_1 x} dH - \int \frac{x}{1 + m_2 x} dH \right)$$

$$= \frac{1}{\lambda} \left( \frac{1}{m_1} - \frac{\lambda}{w} - \frac{1}{m_2} + \frac{\lambda}{2 - w} \right) \qquad \text{(By equation (3.2))} \tag{B.3}$$

$$= \frac{1}{\lambda} \left( f\left(\frac{\lambda}{w}\right) - f\left(\frac{\lambda}{2 - w}\right) \right) \tag{B.4}$$

$$= \frac{1}{\lambda} \left( \int \frac{\lambda}{\lambda + wt} \mu(dt) - \int \frac{\lambda}{\lambda + (2 - w)t} \mu(dt) \right)$$

$$= \int \frac{2(1 - w)t}{(\lambda + wt)(\lambda + (2 - w)t)} \mu(dt).$$

Thus, we obtain that for some measure $\mu$ on $\mathbb{R}^+$, we have

$$\implies \mathcal{V}_\lambda = \int t \frac{w(2 - w)}{(\lambda + wt)(\lambda + (2 - w)t)} \mu(dt) \tag{B.5}$$

Similarly, we can also simplify the bias. If $H = G$, we have from equation (3.9),

$$\mathcal{B}_\lambda = \left( \int \frac{x}{(1 + m_1 x)^2} dH \right) \left( 1 - \gamma \int \frac{m_1^2 x^2}{(1 + m_1 x)^2} dH \right)^{-1}$$

We first multiply both sides of equation (B.1) by $-m_1^2 / m'(-\lambda/w)$ to obtain

$$\gamma \int \frac{m_1^2 x^2}{(1 + m_1 x)^2} dH = 1 - \frac{m_1^2}{m'(-\lambda/w)}.$$

Next, we decompose the first term of the bias as

$$\frac{x}{(1+m_1 x)^2} = \frac{x}{1+m_1 x} - \frac{m_1 x^2}{(1+m_1 x)^2}$$

$$\implies \gamma \int \frac{x}{(1+m_1 x)^2} dH = \gamma \int \frac{x}{1+m_1 x} dH - \gamma \int \frac{m_1 x^2}{(1+m_1 x)^2} dH$$

$$= \frac{1}{m_1} - \frac{\lambda}{w} - \left( \frac{1}{m_1} - \frac{m_1}{m'(\lambda/w)} \right)$$

$$= \frac{m_1}{m'(\lambda/w)} - \frac{\lambda}{w}$$

Combing the two equalities above, we get

$$\mathcal{B}_\lambda = \gamma^{-1} \left( \frac{m_1}{m'(-\lambda/w)} - \frac{\lambda}{w} \right) \frac{m'(-\lambda/w)}{m_1^2}$$

$$= \gamma^{-1} \left( \frac{1}{m(\frac{-\lambda}{w})} - \frac{\lambda}{w} \frac{m'(\frac{-\lambda}{w})}{m(\frac{-\lambda}{w})^2} \right)$$

$$= \gamma^{-1} \left( f(\lambda/w) - \frac{\lambda}{w} f'(\lambda/w) \right) \tag{B.6}$$

Let $z = \lambda/w$, we use equation (B.2) to get

$$\gamma \mathcal{B}_\lambda = a + \int \frac{z}{z+t} - \frac{zt}{(z+t)^2} \mu(dt)$$

$$= a + \int \frac{z^2}{(z+t)^2} \mu(dt)$$

$$= a + \int \frac{\lambda^2}{(\lambda + wt)^2} \mu(dt) \tag{B.7}$$

Since the sum of log-convex functions are log-convex, this implies that $\mathcal{B}_\lambda$ is decreasing and log-convex. Next we show that $c(w)\mathcal{V}_\lambda$ is log-convex. By equation (B.5), we have

$$c(w)\mathcal{V}_\lambda = \int t\gamma \frac{w^2 + (1-w)^2}{(\lambda + wt)(\lambda + (2-w)t)} \mu(dt)$$

The log-convexity of the above expression simply follows from the fact that $w^2 + (1-w)^2$, $(\lambda + wt)^{-1}$ and $(\lambda + (2-w)t)^{-1}$ are all log-convex in $w \in [0,1]$ for all $t, \lambda \geq 0$ and the fact that sums and products of log-convex functions are log convex.

As long as $\mu \neq \delta_0$, we further have that $c(w)\mathcal{V}_\lambda$ is strictly log-convex, a fact crucial for uniqueness of the minimizer of the risk. It can be readily verified from the definition of $f(z)$ that $\mu = \delta_0 \implies m(z) = (a-z)^{-1}$, that is $m$ must be a Stieltjes transform of a degenerate random variable. However, recall that by Remark 3.1, $m$ is the Steiltjes transform of a free convolution between $MP_\gamma$ and $H$. Since $MP_\gamma$ is non-degenerate, we must have that $m(z) \neq (a-z)^{-1}$ and hence the variance is strictly convex.

As long as $\mu(\mathbb{R}^+) > 0$, we further have that $c(w)\mathcal{V}_\lambda$ is strictly log-convex, a fact crucial for uniqueness of the minimizer of the risk. It can be readily verified from the definition of $f(z)$ that $\mu \equiv 0 \implies m(z) = (a-z)^{-1}$, that is $m$ must be a Stieltjes transform of a degenerate random variable. However, recall that by Remark 3.1, $m$ is the Steiltjes transform of a free convolution between $MP_\gamma$ and $H$. Since $MP_\gamma$ is non-degenerate, we must have that $m(z) \neq (a-z)^{-1}$ and hence the variance is strictly convex.

Next, we show that $w^\star$ is a continuous function of $\lambda$. Define the quantity

$$\mathcal{R}(w, \lambda) = \lim_{n \to \infty} \lim_{t \to \infty} R(\widehat{\boldsymbol{\beta}}_{t,\lambda}, \widehat{\boldsymbol{\beta}}).$$

Note that, $\mathcal{R}(w, \lambda)$ is continuous in both $w$ and $\lambda$. Define any convergent sequence $\lambda_n \to \widetilde{\lambda}$. Let $w_k^\star$ to be the unique minimizer of $\mathcal{R}(w, \lambda_k)$ (unique minimizer because $\mathcal{R}$ is strictly log-convex is

all three results of this section). Since $w_n^\star$ is a sequence in the compact set $[0, 1]$, there exists a convergent subsequence $w_{n_k}^\star$ converging to some limit $\widetilde{w}^\star$. For any $\forall w \in [0, 1]$,

$$\mathcal{R}(w_{n_k}, \lambda_{n_k}) \leq \mathcal{R}(w, \lambda_{n_k}) \implies \mathcal{R}(\widetilde{w}, \widetilde{\lambda}) \leq \mathcal{R}(w, \widetilde{\lambda}).$$

That is, $\widetilde{w}^\star$ is the minimizer of $\mathcal{R}(\cdot, \widetilde{\lambda})$. Since the convergent subsequence $n_k$ we picked was arbitrary, we have shown that every convergent subsequence of $w_k$ converges to $\widetilde{w}^\star$ and thus we must have that $w_n^\star \to \widetilde{w}^\star$. This proves that $w^\star$ is a continuous function of $\lambda$.

Next, we propose to show that under $G = H$, the optimal mixing proportion $w^\star$ is in $[1/2, 1]$, with $w^\star(\lambda) \to \phi^{-1}$ as $\lambda \downarrow 0$ and $w^\star(\lambda) \to 1$ as $\lambda \to \infty$. Recall that both $c(w)\mathcal{V}_\lambda$ and $\mathcal{B}_\lambda$ are continuous and log-convex, and $\mathcal{B}_\lambda$ is also decreasing. This immediately tells us that the limiting generalization error is log-convex and hence it has a unique minimizer.

$$\lim_{\lambda \to 0} c(w)\mathcal{V}_\lambda = \frac{w^2 + (1-w)^2}{w(2-w)} \gamma \int \frac{1}{t} \mu(dt) \quad \text{and} \quad \lim_{\lambda \to 0} \mathcal{B}_\lambda = \frac{a}{\gamma} \tag{B.8}$$

The minimizer of the of the risk at $\lambda$ is clearly only dependent on $c(w)$, which is minimized at $\phi^{-1}$. Next, as $\lambda \to \infty$, we have $c(w)\mathcal{V}_\lambda \to 0$. However, since $\mathcal{B}_\lambda$ is a decreasing function of $w$ and $\liminf_{\lambda \to \infty} \mathcal{B}_\lambda > 0$, we must have that $w^\star \to 1$. Finally, we need to show that $w^\star \geq 1/2$. To this end, write the variance as

$$c(w)\mathcal{V}_\lambda = (w^2 + (1-w)^2) \int \frac{t\gamma}{(\lambda + wt)(\lambda + (2-w)t)} \mu(dt)$$

The expression outside of the integral is minimized at $w = 1/2$, while the factor inside the integral is a decreasing function of $w$ for $w \in [0, 1]$. This can be seen by calculating the derivative of the integrand

$$\frac{d}{dw} \frac{1}{(\lambda + wt)(\lambda + (2-w)t)} = \frac{-2t^2(1-w)}{(\lambda + wt)^2(\lambda + (2-w)t)^2} \tag{B.9}$$

Since $\mathcal{B}_\lambda$ is also a decreasing function of $w$, we must have that risk at $w < 1/2$ must be larger than the risk at $1/2$. This completes the proof that $w^\star \geq 1/2$.

PROOF OF PROPOSITION 3.1

We obtain the asymptotic bias and variance for the random effects model from equation (B.6) and equation (B.4) respectively. The log-convexity of variance follows from equation (B.5). $\mathcal{B}_\lambda$ is log-convex and decreasing using equation (B.7). This completes the proof of theorem 3.1.

PROOF OF PROPOSITION 3.2

By Corollary D.1, we obtain that $G = H$. Further, under the assumption $\beta_i \overset{\text{i.i.d.}}{\sim} (0, b_\star/p)$, we obtain $\|\boldsymbol{\beta}\|^2 \to b_\star$ almost surely. Then, by the argument above, we have $w^\star \in [1/2, 1]$, with $w^\star(\lambda) \to \phi^{-1}$ as $\lambda \downarrow 0$ and $w^\star(\lambda) \to 1$ as $\lambda \to \infty$. Finally, to see that $w^\star$ increases with $\text{SNR} = b_\star/\sigma^2$, recall that the risk is $\mathcal{R}(w, b_\star) = \sigma^2 c(w)\mathcal{V}_\lambda + b_\star \mathcal{B}_\lambda$. Note that dividing the risk by $\sigma^2$ does not change does not change $w^\star$. Thus, it is enough to show that $w^\star$ increases with $b_\star$. By the implicit function theorem,

$$\partial_{b_\star} w^\star = -\frac{\partial_{wb_\star}\mathcal{R}}{\partial_{ww}\mathcal{R}} = -\frac{\partial_w \mathcal{B}_\lambda}{\partial_{ww}\mathcal{R}}$$

Since Bias is a decreasing function of $w$ and risk is a strictly convex function of $w$, must have that $\partial_{b_\star} w^\star \geq 0$, thus proving $w^\star$ is a non decreasing function of $b_\star$. 3.2.

PROOF OF THEOREM 3.3

If $\boldsymbol{\Sigma} = \alpha \boldsymbol{I}$, we have $G = H$, and we obtain the desired conclusion by the argument above.

PROOF OF PROPOSITION 3.3

For spike covariance matrix $\Sigma$, we obtain $\widehat{H}_p = p^{-1}\delta_{1+s} + (1 - p^{-1})\delta_1$, where $\delta_x$ is the Dirac measure at the point $x$. Therefore, $\widehat{H}_p \Rightarrow H = \delta_1$. To compute $\widehat{G}_p$, we write the signal $\beta$ as $\beta = \theta v + \sqrt{1 - \theta^2}v^\perp$, where $v^\top v^\perp = 0$ and $\|v^\perp\|_2 = 1$. This implies $\|\beta\|_2 = 1$ and hence $\widehat{G}_p = \theta^2 \delta_{1+s} + (1 - \theta^2)\delta_1$. If $\theta = \theta(n) \to 0$ as $n \to \infty$, we obtain $\widehat{G}_p \Rightarrow H = \delta_1$ and the conclusion of Theorem 3.2 holds. Hence we will restrict ourselves to the case $\theta(n) \to \theta_\star \neq 0$. Here we have $G = \theta_\star^2 \delta_{1+s} + (1 - \theta_\star^2)\delta_1$. Since $H = \delta_1$, the limiting expression of variance is still $\sigma^2 c(w)\mathcal{V}_\lambda$, where $c(w)$ and $\mathcal{V}_\lambda$ are the same as Theorem 3.2.

Turning to the characterization of asymptotic bias, we again use the function $m(z)$ satisfying $m(z)^{-1} + z = \gamma\frac{1}{1+m(z)}$. Defining $m_1 = m(-\lambda/w)$, we have

$$\mathcal{B}_\lambda = \left(\theta_\star^2 \frac{1+s}{(1 + m_1(1+s))^2} + (1 - \theta_\star^2)\frac{1}{(1+m_1)^2}\right)\left(1 - \gamma\frac{m_1^2}{(1+m_1)^2}\right)^{-1}$$

We know from Theorem 3.2 that $\phi_1(m_1) = \frac{1}{(1+m_1)^2 - \gamma m_1^2}$ is a decreasing and log-convex function in $w$. From the above display, we obtain that

$$\mathcal{B}_\lambda = \theta_\star^2(1+s)\underbrace{\frac{(1+m_1)^2}{(1 + m_1(1+s))^2}}_{\phi_2(m_1)}\phi_1(m_1) + (1 - \theta_\star^2)\phi_1(m_1).$$

Since $\phi_2(m_1)$ is also decreasing and log-convex as function of $w$, this implies that $\lim_{n\to\infty}\lim_{t\to\infty} B(\widehat{\beta}_{t,\lambda}; \beta)$ is decreasing and log-convex in $w$. Therefore, asymptotic generalization error has a unique minima $w^\star$. To see the properties of $w^\star$, note that by equation (B.9), we have $w^\star \geq 1/2$. Using equation (B.8), we obtain that $w^\star \to 1/\varphi$ as $\lambda \to 0+$. Finally similar to the proof of Theorem 3.2, we have $\mathcal{V}_\lambda \to 0$ as $\lambda \to \infty$ and $\liminf_{\lambda\to\infty} \mathcal{B}_\lambda > 0$. Since $\mathcal{B}_\lambda$ is decreasing, we again have $\lim_{\lambda\to\infty} w^\star(\lambda) = 1$. This completes the proof of Proposition 3.3.

We conclude the section with with the proof of Lemma B.1.

PROOF OF LEMMA B.1

We need the following definition.

**Definition B.1** (Stieltjes Function ($\mathcal{SF}$))**.** *A function $f : \mathbb{R}^+ \to \mathbb{R}^+$ is called a Stieltjes function if it can be written as*

$$f(x) = \frac{a}{x} + b + \int_{\mathbb{R}^+}\frac{1}{x+t}\mu(dt),$$

*where $a, b \geq 0$ and $\mu$ is a positive measure on $\mathbb{R}^+$ with $\int_{\mathbb{R}^+}\frac{1}{1+t}\mu(dt) < \infty$.*

Recall, the definition $m(z) = \mathbb{E}(A - z)^{-1}$ for some non-negative random variable $A$. Using the definition above, the map $z \to m(-z)$ is a Stieltjes function. Further $\psi(z) = 1/z$ is a Stieltjes function by Definition B.1. Using (Schilling et al., 2009, Theorem 6.2(ii) and Corollary 7.9), we obtain that $1/(zm(-z))$ is a Stieltjes function. Note that $(zm(-z))^{-1} = z^{-1}f(z) + 1$. Hence, by Definition B.1, we have

$$f(z) + z = a + bz + \int\frac{z}{z+t}\mu(dt)$$

with $|f(1)| < \infty$. It remains to show that $b = 1$. This follows from

$$b = \lim_{z\to\infty}\frac{f(z)}{z} = \lim_{z\to\infty}\mathbb{E}[z(A+z)^{-1}]^{-1} = 1$$

where the last equality follows from dominated convergence theorem.

## C  DYNAMIC MIXING

In this Section we prove the following claim for min-$\ell_2$-norm interpolator: Suppose we select the mixing proportion $w_t$ adaptively at each generation to minimize $R(\widehat{\boldsymbol{\beta}}_t; \boldsymbol{\beta})$ for any finite sample size $n$. Then $w_t^\star$ satisfies

$$w_t^\star = \frac{1 + w_{t-1}^\star}{2 + w_{t-1}^\star}, \quad w_0^\star = 1. \tag{C.1}$$

Further, under this setup

$$\lim_{n \to \infty} \lim_{t \to \infty} R(\widehat{\boldsymbol{\beta}}_t; \boldsymbol{\beta}) = \sigma^2 c(w^\star)\mathcal{V} + b_\star \mathcal{B}, \tag{C.2}$$

where $\mathcal{V}, \mathcal{B}$ as in equation (3.5) and $w^\star = 1/\varphi$. To see the claim, note that the bias of $\widehat{\boldsymbol{\beta}}_t$ is independent of mixing proportion, hence it converges to $\mathcal{B}$. Regarding the variance, we will prove the following by induction

$$w_t^\star = \arg\min_w V(\boldsymbol{\beta}_t; \boldsymbol{\beta}), \quad \mathsf{Cov}[\boldsymbol{\beta}_t|\mathbf{X}] = \sigma^2 w_t^\star (\mathbf{X}^\top \mathbf{X})^\dagger. \tag{C.3}$$

Suppose $t = 1$. For any $0 < w_1 < 1$, we have

$$\begin{aligned}
\mathsf{Cov}[\widehat{\boldsymbol{\beta}}_t|\mathbf{X}] &= (\mathbf{X}^\top \mathbf{X})^\dagger \mathbf{X}^\top (w_1^2 \sigma^2 \boldsymbol{I} + 2(1 - w_1)^2 \sigma^2 \mathbf{X}(\mathbf{X}^\top \mathbf{X})^\dagger \mathbf{X}^\top) \mathbf{X}(\mathbf{X}^\top \mathbf{X})^\dagger \\
&= \sigma^2 (w_1^2 + 2(1 - w_1)^2)(\mathbf{X}^\top \mathbf{X})^\dagger.
\end{aligned}$$

Hence the variance is minimized at $w_1^\star = 2/3 = (1 + w_0^\star)/(2 + w_0^\star)$. Further, $w_1^{\star 2} + 2(1 - w_1^\star)^2 = w_1^\star$ proving equation (C.3) for $t = 1$. Now, for any $t > 1$ and any mixing proportion $w_t$, we have by induction hypothesis

$$\begin{aligned}
\mathsf{Cov}[\widehat{\boldsymbol{\beta}}_t|\mathbf{X}] &= (\mathbf{X}^\top \mathbf{X})^\dagger \mathbf{X}^\top (w_t^2 \sigma^2 \boldsymbol{I} + (1 - w_t)^2 \mathsf{Cov}(\widetilde{y}_{t-1}|\mathbf{X})) \mathbf{X}(\mathbf{X}^\top \mathbf{X})^\dagger \\
&= w_t^2 \sigma^2 (\mathbf{X}^\top \mathbf{X})^\dagger + (1 - w_t)^2 (\mathbf{X}^\top \mathbf{X})^\dagger \mathbf{X}^\top \left[ \mathbf{X}\mathsf{Cov}(\widehat{\boldsymbol{\beta}}_{t-1}|\mathbf{X})\mathbf{X}^\top + \sigma^2 \boldsymbol{I} \right] \mathbf{X}(\mathbf{X}^\top \mathbf{X})^\dagger \\
&= (w_t^2 + (1 - w_t)^2(1 + w_{t-1}^\star))\sigma^2 (\mathbf{X}^\top \mathbf{X})^\dagger,
\end{aligned}$$

which is minimized at $w_t^\star = \frac{1 + w_{t-1}^\star}{2 + w_{t-1}^\star}$. Further we have $w_t^{\star 2} + (1 - w_t^\star)^2(1 + w_{t-1}^\star) = w_t^\star$, proving equation (C.3) by induction principle. Since by equation (C.1), we have $w_t^\star \to w^\star$, we immediately have equation (C.2).

## D  RANDOM EFFECTS

**Lemma D.1** (Equal weak limits). *Let $\boldsymbol{\Sigma} = \sum_{k=1}^p s_k \boldsymbol{v}_k \boldsymbol{v}_k^\top$ with eigenvalues $s_1, \ldots, s_p$ and orthonormal eigenvectors $\boldsymbol{v}_1, \ldots, \boldsymbol{v}_p$ Define*

$$\widehat{H}_p(x) = \frac{1}{p} \sum_{k=1}^p \mathbf{1}_{s_k \le x}, \qquad \widehat{G}_p(x) = \frac{1}{\|\boldsymbol{\beta}\|_2^2} \sum_{k=1}^p \langle \boldsymbol{v}_k, \boldsymbol{\beta} \rangle^2 \mathbf{1}_{s_k \le x}.$$

*Assume $\boldsymbol{\beta} = (\beta_1, \ldots, \beta_p)$ has i.i.d. entries with $\mathbb{E}\beta_i = 0$, $\mathbb{E}\beta_i^2 = \tau^2 \in (0, \infty)$ and $\mathbb{E}\beta_i^4 < \infty$, independent of $\boldsymbol{\Sigma}$.*

*Then, conditional on $\boldsymbol{\Sigma}$, for every bounded Lipschitz test function $\psi$,*

$$\int \psi \, d\widehat{G}_p - \int \psi \, d\widehat{H}_p \xrightarrow{\mathbb{P}} 0$$

*as $p \to \infty$. Consequently, if $\widehat{H}_p \Rightarrow H$ weakly, then also $\widehat{G}_p \Rightarrow H$ weakly.*

*Proof.* Fix any bounded Lipschitz continuous function $\psi : \mathbb{R} \to \mathbb{R}$ such that $\|\psi\|_\infty \le M$. Set $a_k := \psi(s_k)$ (which is deterministic given $\boldsymbol{\Sigma}$). Let $\xi_k := \langle \boldsymbol{v}_k, \boldsymbol{\beta} \rangle$. Since $\beta_i$'s are i.i.d. and independent of

$\Sigma$, orthonormality gives that, conditional on $\Sigma$, the variables $\xi_k$ identically distributed and mutually uncorrelated with $\mathbb{E}\xi_k = 0$, $\mathbb{E}\xi_k^2 = \tau^2$, $\mathbb{E}\xi_k^4 < \infty$. Then

$$\int \psi \, d\widehat{G}_p = \frac{\sum_{k=1}^p a_k \xi_k^2}{\sum_{j=1}^p \xi_j^2}, \qquad \int \psi \, d\widehat{H}_p = \frac{1}{p}\sum_{k=1}^p a_k =: A_p.$$

Define centered averages

$$B_p := \frac{1}{p}\sum_{k=1}^p a_k(\xi_k^2 - \tau^2), \qquad C_p := \frac{1}{p}\sum_{j=1}^p (\xi_j^2 - \tau^2).$$

Then

$$\frac{\sum_k a_k \xi_k^2}{\sum_j \xi_j^2} = \frac{p(\tau^2 A_p + B_p)}{p(\tau^2 + C_p)} = A_p + \frac{B_p - A_p C_p}{\tau^2 + C_p}.$$

Since $\psi$ is bounded, $|a_k| \le M$. Using $\mathbb{E}(\xi_1^2) = \tau^2$ and $\mathbb{E}(\xi_1^4) < \infty$,

$$\mathrm{Var}(B_p) = \frac{1}{p^2}\sum_{k=1}^p a_k^2 \mathrm{Var}(\xi_k^2) \le \frac{M^2}{p}\mathrm{Var}(\xi_1^2) = O\left(\frac{1}{p}\right), \qquad \mathrm{Var}(C_p) = \frac{1}{p}\mathrm{Var}(\xi_1^2) = O\left(\frac{1}{p}\right).$$

Hence $B_p = O_{\mathbb{P}}(p^{-1/2})$ and $C_p = O_{\mathbb{P}}(p^{-1/2})$ conditional on $\Sigma$. Also $|A_p| \le M$ and $\tau^2 + C_p \xrightarrow{\mathbb{P}} \tau^2 > 0$. Therefore

$$\left|\int \psi \, d\widehat{G}_p - \int \psi \, d\widehat{H}_p\right| = \left|\frac{B_p - A_p C_p}{\tau^2 + C_p}\right| = O_p\left(\frac{1}{\sqrt{p}}\right) \xrightarrow{\mathbb{P}} 0,$$

again conditional on $\Sigma$. By characterizations of weak convergence, we have that $\widehat{G}_p - \widehat{H}_p \Rightarrow 0$ in probability, and thus any weak limit of $\widehat{H}_p$ is also a weak limit of $\widehat{G}_p$. $\qquad\square$

**Corollary D.1.** *Let $\widehat{H}_p, H, \widehat{G}_p, G$ and $\Sigma$ be as defined in Lemma D.1. Assume $\boldsymbol{\beta} = \eta_p \boldsymbol{\omega}$ where $\eta_p \ne 0$ is some arbitrary sequence of real numbers and $\boldsymbol{\omega} = (\omega_1, \ldots, \omega_p)$ has i.i.d. entries with $\mathbb{E}\omega_i = 0$, $\mathbb{E}\omega_i^2 = \tau^2 \in (0, \infty)$ and $\mathbb{E}\omega_i^4 < \infty$, independent of $\Sigma$.*

*Then, conditional on $\Sigma$, for every bounded Lipschitz test function $\psi$,*

$$\int \psi \, d\widehat{G}_p - \int \psi \, d\widehat{H}_p \xrightarrow{\mathbb{P}} 0$$

*as $p \to \infty$. Consequently, if $\widehat{H}_p \Rightarrow H$ weakly, then also $\widehat{G}_p \Rightarrow H$ weakly.*

*Proof.* Follows from the fact that

$$\widehat{G}_p(x) = \frac{1}{\|\boldsymbol{\beta}\|_2^2}\sum_{k=1}^p \langle \boldsymbol{v}_k, \boldsymbol{\beta}\rangle^2 \, \mathbf{1}_{s_k \le x} = \frac{1}{\|\boldsymbol{\omega}\|_2^2}\sum_{k=1}^p \langle \boldsymbol{v}_k, \boldsymbol{\omega}\rangle^2 \, \mathbf{1}_{s_k \le x} =: \widetilde{G}_p(x),$$

and applying Lemma D.1 on $\widetilde{G}_p$. $\qquad\square$

# E  DETAILS OF SECTION 5

## E.1  ABSENCE OF REAL LABELS

In this subsection, we analyze the following algorithm, where we generate real data once, and then compute the following estimator based on synthetic data generated in each iteration.

Next, we prove Theorem 5.1 which analyzes generalization error for $\widehat{\boldsymbol{\beta}}_t^{(r)}$.

*Proof of Theorem 5.1.* First, we show that, similar to the main manuscript, bias $B(\widehat{\boldsymbol{\beta}}_t^{(r)}, \boldsymbol{\beta})$ does not depend on $w$. To this end, note for $t \ge 1$

$$\mathbb{E}(\widehat{\boldsymbol{\beta}}_1^{(r)}|\mathbf{X}) = (\mathbf{X}^\top \mathbf{X})^\dagger \mathbf{X}^\top\left(w\mathbf{X}\boldsymbol{\beta} + (1-w)\mathbf{X}\mathbb{E}(\widehat{\boldsymbol{\beta}}_0^{(r)}|\mathbf{X})\right) = (\mathbf{X}^\top \mathbf{X})^\dagger \mathbf{X}^\top \mathbf{X}\boldsymbol{\beta},$$

---

**Algorithm 2** Iterative interpolation with no new real data

---

1: **Input:** Dataset $(\mathbf{y}, \mathbf{X})$; mixing proportion $w \in (0, 1)$.
2: **Initialize:** $\widehat{\boldsymbol{\beta}}_0^{(r)} \leftarrow (\mathbf{X}^\top \mathbf{X})^\dagger \mathbf{X}^\top \mathbf{y}$.
3: **for** $t \geq 1$ **do**
4:     Generate synthetic responses: $\widetilde{\mathbf{y}}_t \leftarrow \mathbf{X}\widehat{\boldsymbol{\beta}}_{t-1}^{(r)} + \widetilde{\boldsymbol{\varepsilon}}_t$.
5:     Update estimator:
$$\widehat{\boldsymbol{\beta}}_t^{(r)} \leftarrow (\mathbf{X}^\top \mathbf{X})^\dagger \mathbf{X}^\top \big(w\mathbf{y}_0 + (1-w)\widetilde{\mathbf{y}}_t\big).$$
6: **end for**

---

using the definition of $\widehat{\boldsymbol{\beta}}_0^{(r)}$ in Algorithm 2. By induction, $\mathbb{E}(\widehat{\boldsymbol{\beta}}_t^{(r)}|\mathbf{X}) = (\mathbf{X}^\top \mathbf{X})^\dagger \mathbf{X}^\top \big(w\mathbf{X}\boldsymbol{\beta} + (1 - w)\mathbf{X}\mathbb{E}(\widehat{\boldsymbol{\beta}}_{t-1}^{(r)}|\mathbf{X})\big) = (\mathbf{X}^\top \mathbf{X})^\dagger \mathbf{X}^\top \mathbf{X}\boldsymbol{\beta}$, yielding our conclusion that the bias converges similar to the proof of Theorem 3.1. To compute the variance, we will inductively show that for $t \geq 1$,

$$\widehat{\boldsymbol{\beta}}_t^{(r)} = (\mathbf{X}^\top \mathbf{X})^\dagger \mathbf{X}^\top \left( \mathbf{y}_0 + \sum_{j=1}^t (1-w)^{t+1-j}\widetilde{\boldsymbol{\varepsilon}}_j \right). \tag{E.1}$$

If $t = 1$, we have

$$\begin{aligned}
\widehat{\boldsymbol{\beta}}_1^{(r)} &= (\mathbf{X}^\top \mathbf{X})^\dagger \mathbf{X}^\top \big(w\mathbf{y}_0 + (1-w)\widetilde{\mathbf{y}}_1\big) = (\mathbf{X}^\top \mathbf{X})^\dagger \mathbf{X}^\top \big(w\mathbf{y}_0 + (1-w)\mathbf{X}\widehat{\boldsymbol{\beta}}_0^{(r)} + (1-w)\widetilde{\boldsymbol{\varepsilon}}_1\big) \\
&= (\mathbf{X}^\top \mathbf{X})^\dagger \mathbf{X}^\top \big(w\mathbf{y}_0 + (1-w)\mathbf{X}(\mathbf{X}^\top \mathbf{X})^\dagger \mathbf{X}^\top \mathbf{y}_0 + (1-w)\widetilde{\boldsymbol{\varepsilon}}_1\big) \\
&= (\mathbf{X}^\top \mathbf{X})^\dagger \mathbf{X}^\top \big(\mathbf{y}_0 + (1-w)\widetilde{\boldsymbol{\varepsilon}}_1\big).
\end{aligned}$$

This proves equation (E.1) for $t = 1$. Suppose the claim is true for general $t$. Then

$$\begin{aligned}
\widehat{\boldsymbol{\beta}}_{t+1}^{(r)} &= (\mathbf{X}^\top \mathbf{X})^\dagger \mathbf{X}^\top \big(w\mathbf{y}_0 + (1-w)\widetilde{\mathbf{y}}_{t+1}\big) \\
&= (\mathbf{X}^\top \mathbf{X})^\dagger \mathbf{X}^\top \big(w\mathbf{y}_0 + (1-w)\mathbf{X}\widehat{\boldsymbol{\beta}}_t^{(r)} + (1-w)\widetilde{\boldsymbol{\varepsilon}}_{t+1}\big) \\
&= (\mathbf{X}^\top \mathbf{X})^\dagger \mathbf{X}^\top \left( w\mathbf{y}_0 + (1-w)\mathbf{X}(\mathbf{X}^\top \mathbf{X})^\dagger \mathbf{X}^\top \left( \mathbf{y}_0 + \sum_{j=1}^t (1-w)^{t+1-j}\widetilde{\boldsymbol{\varepsilon}}_j \right) + (1-w)\widetilde{\boldsymbol{\varepsilon}}_{t+1} \right) \\
&= (\mathbf{X}^\top \mathbf{X})^\dagger \mathbf{X}^\top \left( \mathbf{y}_0 + \sum_{j=1}^{t+1} (1-w)^{t+2-j}\widetilde{\boldsymbol{\varepsilon}}_j \right),
\end{aligned}$$

proving our claim. This further yields that

$$\begin{aligned}
\lim_{t\to\infty} \mathrm{Tr}(\mathsf{Cov}(\widehat{\boldsymbol{\beta}}_t^{(r)}|\mathbf{X})\boldsymbol{\Sigma}) &= \sigma^2 \left( \sum_{j=0}^\infty (1-w)^2 \right) \mathrm{Tr}\left[(\mathbf{X}^\top \mathbf{X})^\dagger \boldsymbol{\Sigma}\right] \\
&= \frac{\sigma^2}{w(2-w)} \mathrm{Tr}\left[(\mathbf{X}^\top \mathbf{X})^\dagger \boldsymbol{\Sigma}\right].
\end{aligned}$$

Hence, the risk is minimized at $w^\star = 1$ and the quantity $\lim_{n\to\infty}\lim_{t\to\infty} V(\widehat{\boldsymbol{\beta}}_t^{(r)}, \boldsymbol{\beta})$ can be obtained similar to main manuscript. $\qquad\square$

### E.2    TIME-VARYING COVARIATES

In this section, we want to consider the setup where $\mathbf{X}_t$ is generated independently for each $t$, with $\boldsymbol{\Sigma} = \boldsymbol{I}$. Formally, we consider the estimator $\widehat{\boldsymbol{\beta}}_t^{(v)}$ and we want to characterize, for each $t$,

$$B_t := \lim_{n\to\infty} B(\widehat{\boldsymbol{\beta}}_t^{(v)}; \boldsymbol{\beta}), \qquad V_t := \lim_{n\to\infty} V(\widehat{\boldsymbol{\beta}}_t^{(v)}; \boldsymbol{\beta}).$$

---

**Algorithm 3** Time-varying covariates

---

1: **Input:** Dataset $(\mathbf{y}, \mathbf{X})$; mixing proportion $w \in (0, 1)$.

2: **Initialize:** $\widehat{\boldsymbol{\beta}}_0^{(v)} \leftarrow (\mathbf{X}^\top \mathbf{X})^\dagger \mathbf{X}^\top \mathbf{y}$.

3: **for** $t \geq 1$ **do**

4:     Generate real responses: $\mathbf{y}_t \leftarrow \mathbf{X}_t \boldsymbol{\beta} + \varepsilon_t$.

5:     Generate synthetic responses: $\widetilde{\mathbf{y}}_t \leftarrow \mathbf{X}_t \widehat{\boldsymbol{\beta}}_{t-1}^{(v)} + \widetilde{\varepsilon}_t$.

6:     Update estimator:
$$\widehat{\boldsymbol{\beta}}_t^{(v)} \leftarrow (\mathbf{X}_t^\top \mathbf{X}_t)^\dagger \mathbf{X}_t^\top \big( w \mathbf{y}_t + (1 - w) \widetilde{\mathbf{y}}_t \big).$$

7: **end for**

---

**Remark E.1.** *Note that in both Theorem E.1 and subsequent Theorem E.2, we analyze the asymptotic risk in the order* $\lim_{t \to \infty} \lim_{n \to \infty} R(\widehat{\boldsymbol{\beta}}_t^{(v)}; \boldsymbol{\beta})$, *rather than* $\lim_{n \to \infty} \lim_{t \to \infty} R(\widehat{\boldsymbol{\beta}}_t^{(v)}; \boldsymbol{\beta})$ *used in the main manuscript. This ordering is necessary because the covariates are generated independently at each iteration in Theorem E.1 ($\mathbf{X}_t$) and Theorem E.2 ($\mathbf{X}_t, \widetilde{\mathbf{X}}_t$). Consequently, for each fixed iteration $t$, the risk must first be averaged over the randomness of the newly generated covariates as $n \to \infty$.*

After the proof of our result, we provide two simulations to complement our findings. Figure 4a plots optimal mixing parameter $w^\star$ as a function of $\gamma$. Our plot shows, unlike Theorem 3.1, $w^\star$ depends on $\gamma$. Further, Figure 4b shows empirical risk matches with our theoretical prediction in moderate sample size.

**Theorem E.1.** *Assume* $\widehat{\boldsymbol{\beta}}_t^{(v)}$ *is obtained via Algorithm 3 and* $\boldsymbol{\Sigma} = \boldsymbol{I}$. *Then we have*

$$\lim_{t \to \infty} B_t = \frac{b_\star \left( 1 - \frac{1}{\gamma} \right)}{1 - \frac{1}{\gamma}(1 - w)^2}, \qquad \lim_{t \to \infty} V_t = \frac{(w^2 + (1 - w)^2)\sigma^2}{(\gamma - 1)\left( 1 - \frac{1}{\gamma}(1 - w)^2 \right)}.$$

*Hence, risk is optimized at* $w^\star = \frac{1}{2} \left( \sqrt{(c + 2\gamma)^2 + 1 + 2c} - (c + 2\gamma - 1) \right)$ *where* $c = \frac{b_\star}{\sigma^2}(\gamma - 1)^2 \gamma^{-1}$

*Proof.* We begin by characterizing the bias $B(\widehat{\boldsymbol{\beta}}_t^{(v)}; \boldsymbol{\beta})$ conditioned on $\mathbf{X}_{1:t} = (\mathbf{X}_j)_{1 \leq j \leq t}$:

$B(\widehat{\boldsymbol{\beta}}_t^{(v)}; \boldsymbol{\beta}) = \|\boldsymbol{\beta} - \mathbb{E}[\widehat{\boldsymbol{\beta}}_t^{(v)} | \mathbf{X}_{1:t}]\|_2^2$

$= \|\boldsymbol{\beta} - (\mathbf{X}_t^\top \mathbf{X}_t)^\dagger \mathbf{X}_t^\top \mathbf{X}_t \big( w \boldsymbol{\beta} + (1 - w) \mathbb{E}[\widehat{\boldsymbol{\beta}}_{t-1}^{(v)} | \mathbf{X}_{1:(t-1)}] \big)\|_2^2$

$= \|(I - (\mathbf{X}_t^\top \mathbf{X}_t)^\dagger \mathbf{X}_t^\top \mathbf{X}_t)\boldsymbol{\beta} + (1 - w)(\mathbf{X}_t^\top \mathbf{X}_t)^\dagger \mathbf{X}_t^\top \mathbf{X}_t (\boldsymbol{\beta} - \mathbb{E}[\widehat{\boldsymbol{\beta}}_{t-1}^{(v)} | \mathbf{X}_{1:(t-1)}])\|_2^2$

$= \boldsymbol{\beta}^\top (I - (\mathbf{X}_t^\top \mathbf{X}_t)^\dagger \mathbf{X}_t^\top \mathbf{X}_t)\boldsymbol{\beta} + (1 - w)^2 (\boldsymbol{\beta} - \mathbb{E}[\widehat{\boldsymbol{\beta}}_{t-1}^{(v)} | \mathbf{X}_{1:(t-1)}])^\top (\mathbf{X}_t^\top \mathbf{X}_t)^\dagger \mathbf{X}_t^\top \mathbf{X}_t (\boldsymbol{\beta} - \mathbb{E}[\widehat{\boldsymbol{\beta}}_{t-1}^{(v)} | \mathbf{X}_{1:(t-1)}])$

Therefore, conditioned on $\mathbf{X}_{1:(t-1)}$, we obtain the high probability limits of the two summands above separately. For the first summand, note that (Hastie et al., 2022, (114)) that

$$\boldsymbol{\beta}^\top (I - (\mathbf{X}_t^\top \mathbf{X}_t)^\dagger \mathbf{X}_t^\top \mathbf{X}_t)\boldsymbol{\beta} \to b_\star \left( 1 - \frac{1}{\gamma} \right),$$

with high probability where $b_\star = \lim \|\boldsymbol{\beta}\|^2$. Similarly, the second summand converges with high probability, to

$$\frac{1}{\gamma}(1 - w)^2 \|(\boldsymbol{\beta} - \mathbb{E}[\widehat{\boldsymbol{\beta}}_{t-1}^{(v)} | \mathbf{X}_{1:(t-1)}])\|_2^2.$$

Hence, we obtain that $B_t = \lim_{n \to \infty} B(\widehat{\boldsymbol{\beta}}_t^{(v)}; \boldsymbol{\beta})$ satisfies the recursion

$$B_t = b_\star \left( 1 - \frac{1}{\gamma} \right) + \frac{1}{\gamma}(1 - w)^2 B_{t-1}.$$

Since $\frac{1}{\gamma}(1-w)^2 < 1$, the sequence converges to $\frac{b_\star\left(1-\frac{1}{\gamma}\right)}{1-\frac{1}{\gamma}(1-w)^2}$ as $t \to \infty$.

Next, we characterize the variance $V(\widehat{\boldsymbol{\beta}}_t^{(v)}; \boldsymbol{\beta})$ conditioned on $\mathbf{X}_{1:t} = (\mathbf{X}_j)_{1 \le j \le t}$. To this end, note that,

$$V(\widehat{\boldsymbol{\beta}}_t^{(v)}; \boldsymbol{\beta}) = \text{Tr}[\text{Cov}(\widehat{\boldsymbol{\beta}}_t^{(v)}|\mathbf{X}_{1:t})]$$

$$= w^2\sigma^2\text{Tr}[(\mathbf{X}_t^\top \mathbf{X}_t)^\dagger] + (1-w)^2\text{Tr}[(\mathbf{X}_t^\top \mathbf{X}_t)^\dagger \mathbf{X}_t^\top \left\{ \mathbf{X}_t\text{Cov}(\widehat{\boldsymbol{\beta}}_{t-1}^{(v)}|\mathbf{X}_{1:(t-1)})\mathbf{X}_t^\top + \sigma^2\boldsymbol{I} \right\} \mathbf{X}_t(\mathbf{X}_t^\top \mathbf{X}_t)^\dagger]$$

$$= (w^2 + (1-w)^2)\sigma^2\text{Tr}[(\mathbf{X}_t^\top \mathbf{X}_t)^\dagger] + (1-w)^2\text{Tr}[(\mathbf{X}_t^\top \mathbf{X}_t)^\dagger(\mathbf{X}_t^\top \mathbf{X}_t)\text{Cov}(\widehat{\boldsymbol{\beta}}_{t-1}^{(v)}|\mathbf{X}_{1:(t-1)})]$$

Recalling that $V_t = \lim_{n \to \infty} V(\widehat{\boldsymbol{\beta}}_t^{(v)}; \boldsymbol{\beta})$, we have, by argument similar to above, that

$$V_t = \frac{1}{\gamma - 1}(w^2 + (1-w)^2)\sigma^2 + (1-w)^2\frac{1}{\gamma}V_{t-1}.$$

Therefore, we obtain $V_t$ converges to $\frac{(w^2+(1-w)^2)\sigma^2}{(\gamma-1)\left(1-\frac{1}{\gamma}(1-w)^2\right)}$ as $t \to \infty$. □

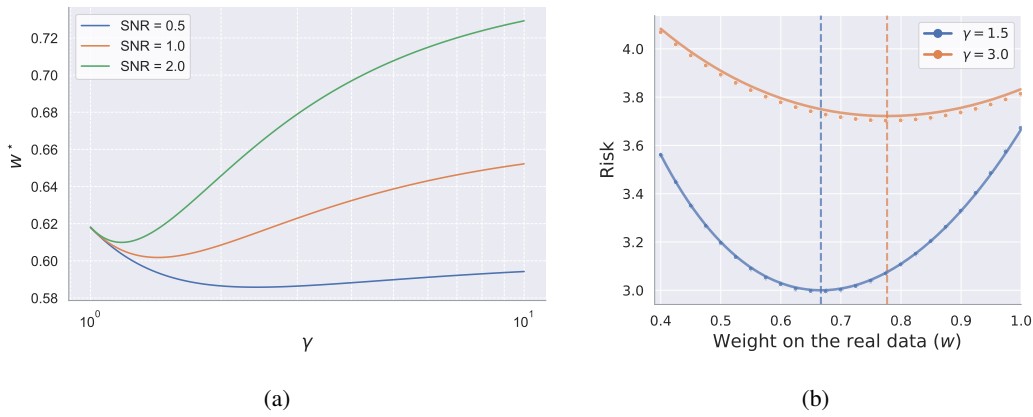

(a)                  (b)

Figure 4: Panel $(a)$: We plot theoretical values of optimal mixing parameter $w^\star$ as a function of $\gamma$ as obtained in Theorem E.1. We vary $\text{SNR} = \{0.5, 1, 2\}$. Our plots show that $w^\star$ is non-monotone as function of $\gamma$. Panel $(b)$: We empirically verify the generalization error of Theorem E.1. For $\gamma = \{1.5, 3\}$ we plot risk as a function of $w$. The dotted lines correspond to empirical risk and the solid lines represent theoretical risk. The vertical dashed lines correspond to optimal mixing value $w^\star$. The empirical risk is computed with $\text{SNR} = 5$ and $n = 1500$, iteration $t = 20$.

### E.3 CONCATENATION OF REAL AND SYNTHETIC DATA

---

**Algorithm 4** Concatenation in online setup

---

1: **Input:** Dataset $(\mathbf{y}, \mathbf{X})$; mixing proportion $w \in (0, 1)$.

2: **Initialize:** $\widehat{\boldsymbol{\beta}}_0^{(a)} \leftarrow (\mathbf{X}^\top\mathbf{X})^\dagger\mathbf{X}^\top\mathbf{y}$.

3: **for** $t \ge 1$ **do**

4:     Generate real responses: $\mathbf{y}_t \leftarrow \mathbf{X}_t\boldsymbol{\beta} + \boldsymbol{\varepsilon}_t$.

5:     Generate synthetic responses: $\widetilde{\mathbf{y}}_t \leftarrow \widetilde{\mathbf{X}}_t\widehat{\boldsymbol{\beta}}_{t-1}^{(a)} + \widetilde{\boldsymbol{\varepsilon}}_t$.

6:     Update estimator:
$$\widehat{\boldsymbol{\beta}}_t^{(a)} \leftarrow (\mathbf{X}_t^\top\mathbf{X}_t + \widetilde{\mathbf{X}}_t^\top\widetilde{\mathbf{X}}_t)^\dagger(\mathbf{X}_t^\top\mathbf{y}_t + \widetilde{\mathbf{X}}_t^\top\widetilde{\mathbf{y}}_t).$$

7: **end for**

---

Here, we consider the setup where $\mathbf{X}_t$, $\widetilde{\mathbf{X}}_t$ both are Gaussian data matrix with $\mathbf{\Sigma} = \mathbf{I}$. We want to analyze whether model collapse can be prevented by constructing an estimator by pooling the real and synthetic labels as delineated by Algorithm 4. We have the following result which shows for certain regimes of overparametrization, model collapse might not be preventable. Figure 5 indeed shows that the asymptotic generalization error diverges.

**Theorem E.2.** *Assume $\widehat{\boldsymbol{\beta}}_t^{(a)}$ is obtained via Algorithm 4, $\gamma > 2$ and $\mathbf{\Sigma} = I$. If $\frac{(\gamma-1)}{\gamma(\gamma-2)} > 1$, then*

$$\lim_{t\to\infty} \lim_{n\to\infty} R(\widehat{\boldsymbol{\beta}}_t^{(a)}; \boldsymbol{\beta}) = \infty$$

*Proof.* We begin by characterizing the bias $B_t = \lim_{n\to\infty} B(\widehat{\boldsymbol{\beta}}_t^{(a)}, \boldsymbol{\beta})$, and $R_t = \lim_{n\to\infty} R(\widehat{\boldsymbol{\beta}}_t^{(a)}, \boldsymbol{\beta})$ where we compute the bias and variance conditioned on $\mathbf{X}_{1:t} = (\mathbf{X}_j, \widetilde{\mathbf{X}}_j)_{1 \le j \le t}$. Note that, to define the estimation $\widehat{\boldsymbol{\beta}}_t$ in Algorithm 4, we need $p > 2n$, hence we will consider $\gamma > 2$. For any finite $n$, $B(\widehat{\boldsymbol{\beta}}_t^{(a)}, \boldsymbol{\beta})$ satisfies the decomposition (Song et al., 2024, (2.13)):

$$B(\widehat{\boldsymbol{\beta}}_t^{(a)}; \boldsymbol{\beta}) = \underbrace{\boldsymbol{\beta}^\top (\widehat{\mathbf{\Sigma}}^\dagger \widehat{\mathbf{\Sigma}} - \boldsymbol{I})^2 \boldsymbol{\beta}}_{B_1(\widehat{\boldsymbol{\beta}}_t; \boldsymbol{\beta})} + \underbrace{\widetilde{\boldsymbol{\beta}}_{t-1}^\top \Big(\frac{\widetilde{\mathbf{X}}_t^\top \widetilde{\mathbf{X}}_t}{2n}\Big) \big(\widehat{\mathbf{\Sigma}}_t^\dagger\big)^2 \Big(\frac{\widetilde{\mathbf{X}}_t^\top \widetilde{\mathbf{X}}_t}{2n}\Big) \widetilde{\boldsymbol{\beta}}_{t-1}}_{B_2(\widehat{\boldsymbol{\beta}}_t; \boldsymbol{\beta})}, \qquad (\text{E.2})$$

where $\widetilde{\boldsymbol{\beta}}_{t-1} := \boldsymbol{\beta} - \widehat{\boldsymbol{\beta}}_{t-1}$, $\widehat{\mathbf{\Sigma}}_t = \frac{1}{2n}(\mathbf{X}_t^\top \mathbf{X}_t + \widetilde{\mathbf{X}}_t^\top \widetilde{\mathbf{X}}_t)$. Using (Song et al., 2024, Theorem 3.1), we obtain with high probability,

$$B_t \ge \frac{(\gamma - 1)}{\gamma(\gamma - 2)} B_{t-1}, \qquad (\text{E.3})$$

Therefore, if $\frac{(\gamma-1)}{\gamma(\gamma-2)} > 1$, we obtain that $R_t \to \infty$ as $t \to \infty$. This completes the proof of the Theorem. $\square$

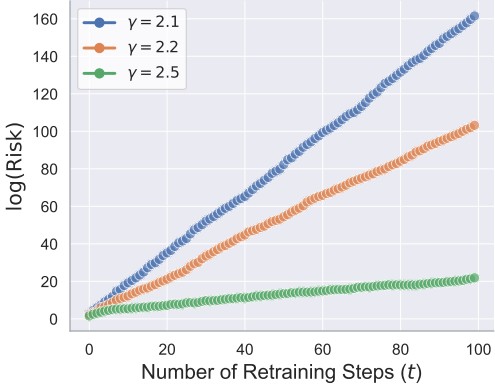

Figure 5: We plot generalization error of $\widehat{\beta}_t^{(a)}$ to complement theoretical findings of Theorem E.2. The dotted lines correspond to $\log(\text{Risk})$ with SNR $= 1$, and $n = 100$ and different values of $\gamma$. The $x$-axis corresponds to number of iterations $t$, which means risk increases exponentially with iterations and resulting in model collapse.

