# OpenReview forum: "Preventing Model Collapse Under Overparametrization: Optimal Mixing Ratios for Interpolation Learning and Ridge Regression"
_ICLR.cc/2026/Conference — ICLR 2026 Poster_

### Official Review · Reviewer_LCBS · 2025-10-17

**Soundness:** 3
**Presentation:** 2
**Contribution:** 2
**Rating:** 4
**Confidence:** 5

**Summary:**

This paper mainly studies how to prevent model collapse in a setting where in each iteration, a batch of new real data and a batch of synthetic data from the previous round are used to train the next round's generative model. The paper first studies this problem for overparametrized linear regression, which seems to be an extension of a previous paper.

Next, the ridge regression is also analyzed for two popular model classes:  random-effects model and the spiked covariance model.

**Strengths:**

The paper is clearly written and easy to understand.

Most results presented in this paper are sound.

Theorems in this paper are supported by both theoretical proofs and experiments.

**Weaknesses:**

The first main concern for this paper is that the assumptions are too idealized, making it difficult for the results to be meaningful in practice. To be more specific, this paper mainly considers standard linear regression and ridge regression  in a **fixed design setting**. That is, the design matrix X is fixed across iterations. This setting is unrealistic in practice, as when people retrain a language model using synthetic data, they typically do not keep the prompts fixed across iterations. Also, this fixed design setting makes the ratio of the synthetic data and real data to be 1 in each iteration, which seems to be rare in practice.

Second, the contribution in this paper seems to be incremental. For example, if my understanding is correct, both this paper and the earlier paper cited by this paper(https://arxiv.org/pdf/2502.18049) aim to study how to use weighting to prevent model collapse. Theorem 3.1 in this paper aims to study this for overparametrized linear regression, but when I read the proof, it seems the weighting strategy can be derived from Lemma A.2 in the appendix directly, and from Lemma A.2 we can see that the weighting strategy for linear regression does not depend on the dimension. Therefore, it is unclear to me what important new insight can be brought by this section.

Also, there exists many typos in the paper that need to be fixed. For example, in appendix C, “Further, $w_1^{\star 2} + (1 - w_1^{\star})^2 = w_1^{\star}.$”  should be "Further, $w_1^{\star 2} + 2(1 - w_1^{\star})^2 = w_1^{\star}.$". In (C.3), the term $\Sigma$ should not appear.
In the proof of Lemma B.1, (appendix page 22), the identity
$\frac{1}{zm(-z)} = \frac{f(z)}{z} - 1$
appears to be incorrect.  The term "-1" should be replaced by "+1".
In equation (B.4), $ \frac{\lambda}{2}-w$ should be corrected to $\frac{\lambda}{2-w}$.

Moreover, in the proof for proposition 3.3, the authors seem to assume $\lim_{ n \to \infty}$ and $\lim_{t \to \infty}$ can be exchanged. In the main text, it takes the limit
with respect to $n$ first (i.e., $\lim_{n\to\infty}\lim_{t\to\infty}$),
but in the proof in the appendix the analysis is carried out in the opposite order
(i.e., $\lim_{t\to\infty}\lim_{n\to\infty}$) without justification of their interchangeability.

**Questions:**

Please address the issues mentioned in the weaknesses section. If at least some of them are addressed, or even the typos are fixed, I would be happy to raise my score.

---

> ### Author Response · Authors · 2025-11-20
> **Response to Reviewer LCBS**
>
> # Response to weakness 1
> Thank you for the insightful thoughts. To address your concerns, we have added three new settings in Section 5 (details in Appendix E). Please see the common official comment posted above where we describe our new contributions in detail. To summarize, among these new settings, both the setting "Time-varying covariates" and "Time-varying covariates with fresh labels in a fraction" allow the covariates to change over time. In the former, the remaining setting is the same as in our original submission. We characterize the risk and optimal mixing weight in Theorem E.1, Appendix E.2. In the latter, we consider the covariates to change over time and additionally, at each iteration, only a fraction of covariates get to observe a real label. In this case, we find a range of $\gamma$ values where model collapse is inevitable (Theorem E.2, Appendix E.3).
>
> # Response to weakness 2
>  Thank you for the comment, this is not true for ridge regression. As explained also in the common official comment above, Lemma A.2 applies only to the minimum norm interpolator. A major contribution of our paper is to understand properties of ridge regression under mixing of real and synthetic data when the number of variables is large (Sections 3.2-3.4). This component relies heavily on being able to provide neat limiting formulae for the risk---a step that is non-trivial and note, we are able to establish this for quite general settings (see Section 3.3-3.4). In  Theorem 3.3, we showed the dependence of the optimal mixing ratio for ridge on $\lambda$ and the SNR. However, the optimal mixing ratio is not constant with respect to $\gamma$, but the relationship is not as direct as that with the other two.  Figure 3(a) now demonstrates the relationship between the optimal mixing weight and $\gamma$. On examining this plot, the effect of dimensionality becomes clear.  Unlike us, [1] focuses on GLMs in low dimensions, which do not exhibit any dimensionality effect, and study maximum likelihood in that context.  Thus, our entire contribution on ridge theory in high dimensions is new in the model collapse literature, and we do not believe this is incremental.
>
> # Response to weakness 3
>  Thank you so much for spotting these. We have fixed these and others we found with a thorough pass, but please let us know if you notice anything that we missed accidentally.
>
> # Response to questions
>  Our extensive new Section 5 (and Appendix E) address the main questions you raised. We thank you once again for the insightful review.
>
> [1] He et al. "Golden ratio mixing of real and synthetic data for stabilizing generative model training."

---

> ### Comment · Reviewer_LCBS · 2025-11-20
>
> Thank you for the rebuttal. **I have increased my score to 6, as most issues seem being addressed.**
>
> However, in the current version, your manuscript states:
>
> ``This phenomenon was previously proved only for ordinary least squares in low-dimensional linear regression with Gaussian features \cite{he2025}.''
>
> After carefully checking \cite{he2025}, I noticed that their results seem broader than the above sentences suggest. In particular, they also discuss generalized linear models under some appropriate regularity assumptions and they also talk about a nonparametric model.
>
> I therefore recommend revising the sentences to more accurately reflect the scope of their results, as in your rebuttal you explicitly acknowledge that \cite{he2025} does consider the GLM setting. It would be helpful to clarify this difference between what is written in the manuscript and what is stated in the rebuttal, and to update the description of this related work accordingly.
>
> In addition, your rebuttal refers to a work titled ``Golden ratio mixing of real and synthetic data for stabilizing generative model training.'' I searched for this title and it seems to correspond to an earlier version of He et al.'s work. Could you clarify which specific paper this refers to and add the appropriate reference?
>
> **Although I have raised my score, please more clearly explain the contributions from the prior work; doing so will help our machine learning theory community better understand how these results connect and how the field is progressing.**

---

> > ### Author Response · Authors · 2025-11-20
> > **Fixed mentioned inconsistencies (For reviewer LCBS)**
> >
> > Thank you for pointing this out. We completely agree that this is important so the community knows better how these results connect and where the field is headed. Indeed the Oct, 2025 updated version on arXiv of He et al. studies settings beyond the linear model which we pointed to in the response letter, but missed updating in the paper. We have done this now. Given that the prior issues raised have been addressed, and we expanded the literature to make things accurate and detailed, we are wondering if you might consider revisiting your current score to an accept rather than marginally accept (we have really put in efforts to address all major comments from the reviewers, as you can see in the new Section 5). But of course, we are respectful to whatever your final decision may be. Once again, thank you for the thorough reading of our paper and for your   comments that helped improve the paper.

---

> > > ### Comment · Reviewer_LCBS · 2025-11-20
> > >
> > > Thank you for the rebuttal. Now I think most of my concerns have been addressed, and the paper now presents a solid theoretical result.
> > >
> > > Therefore, I have updated my score to 8.
> > >
> > > Good luck with the remainder of the rebuttal.

---

### Official Review · Reviewer_Fx5M · 2025-10-30

**Soundness:** 4
**Presentation:** 3
**Contribution:** 2
**Rating:** 4
**Confidence:** 4

**Summary:**

The paper presents a theoretical analysis of model collapse in overparameterized linear models in the ridge and ridgeless regression regime. The paper considers the fresh data augmentation framework with fixed covariates and fresh labels at each iteration. Interestingly, the authors identify optimal mixing ratios that minimize prediction error in this context which demonstrate how mixing real-data with synthetic outputs mitigates model collapse, notably with the appearance of the golden ratio. Specifically, the authors consider a fixed design matrix $X$ with i.i.d. rows in the proportional regime, and repeatedly refit on responses $ w y_t+(1-w) \tilde{y}_t$. The paper considers the limit of $n/p \to \gamma > 1$ when $t \to \infty$ and derives the exact asymptotic risk as a function of $w$.

**Strengths:**

* The paper is well-organized and clearly written. The theory is sound as far as I checked.
* The paper provides high dimensional generalization of the conclusion in He et al. (2025), showing the same optimal mixing ratio of $\varphi^{-1}$ persists in high dimension.
* The technical result of the paper is fairly generic, covering general $\Sigma$ structure and ridge regression with penalty $\lambda$. The paper also provides simulations that visualize the theoretical claims.

He, Hengzhi, Shirong Xu, and Guang Cheng. "Golden ratio weighting prevents model collapse." arXiv preprint arXiv:2502.18049 (2025).

**Weaknesses:**

Major comments:

* The model setup is somewhat too simplistic. The paper essentially assumes the following: (i) infinite supply of fresh labels (ii) the covariate matrix is fixed, and at each data point $X$, it is guaranteed to couple with (infinitely many) observations $y$. A very natural question in this setup is then: why don't we remember all new labels $y_t$, and simply taking $\bar y_t = \sum_{i=1}^t y_i / t$ , and solve the min-norm interpolator on the data $(X, \bar{y}_t)$? This will completely get rid of the noise term and leave only the bias, which is strictly better than any mixing estimator with synthetic data.
* The conclusion of model collapse is a bit narrow. It seems as long as $w > 0$, the estimator will always be reasonable (albeit, suboptimal) even if $t \to \infty$. This does not quite support the empirical evidence in Shumailov et al. (2024) where there is an actual degeneration of the learned distribution.
* The major weakness of the paper is that the contributions are more technical rather than insightful: it provides limiting risks of mixing estimators in high dimensions under general $\Sigma$ and $\lambda$, but the setup is essentially the same as is discussed in He et al. (2025). For example in Eq. (A.5) and Appendix C, the nonasymptotic formula without making use of Marchenko-Pastur limits is sufficient to see the optimal $w$---the conclusion of optimal mixing ratio is not a high dimensional phenomenon, but only the limiting risk formulae. I feel the limiting risk formulae themselves provide little insights about model collapse.

Shumailov, Ilia, et al. "The Curse of Recursion: Training on Generated Data Makes Models Forget." CoRR (2023).
He, Hengzhi, Shirong Xu, and Guang Cheng. "Golden ratio weighting prevents model collapse." arXiv preprint arXiv:2502.18049 (2025).


Minor comments:
* It could be helpful in the statement of Theorem 3.3 showing the numerical value of $\phi^{-1} \approx 0.618$ so the presentation is more clear. Although it was pointed out in P5, Line 235, but I think it still helps the presentation.

**Questions:**

* My main questions are in the weaknesses section. The following are additional questions that I think the paper could incorporate to discuss. Overall I think the paper is technically sound, but the contributions in high dimension is somewhat tangential to the main theme of model collapse.
* What do the authors think of model collapse if fresh labels are only provided in a selected portions of covariates? It is more natural to make use of unlabeled covariates by synthetic labeling. How would this perform in an online setup?
* I appreciate the numerical simulations that unpack the mathematical formulae---however, how would those connect to empirical findings in model collapse? I would like to see more discussions.
* The general $\Sigma$ with bounded spectra is essentially a corollary of the isotropic case. Would the authors comment on model collapse in the scaling law setups? Namely the spectra decay to 0 with infinite number of features such as in Cheng & Montanari (2024).

Cheng, Chen, and Andrea Montanari. "Dimension free ridge regression." The Annals of Statistics 52.6 (2024): 2879-2912.

---

> ### Author Response · Authors · 2025-11-20
> **Response to Reviewer Fx5M**
>
> # Response to Weakness 1 (W1)
> We address the first two concerns in the newly added Section 5. In the first part, we study a setting where the real data is observed once, and synthetic data at each iteration is generated using the previous model. Here the optimal weight becomes 1, implying that adding synthetic data does not improve performance. To address the second comment, we add the ``Time-varying covariates'' part in Section 5, where the covariates change at each iteration. We derive the generalization error and optimal mixing weight in Appendix E.2, Theorem E.1. We describe these in further detail in the common official comment.
> Regarding summing previous labels, this can perform better than our estimator. However, in many cases, old data must be deleted due to legal regulations [2] or become useless due to temporal drift [3]. So we do not pursue this. Among prior work mixing real and synthetic data, our paper is motivated by [1] that empirically shows that lack of fresh data augmentation causes model collapse.  Thus, we study this interesting case where model collapse is NOT inevitable, focusing on identifying weighting mechanisms that not only prevent model collapse  but also improve the original estimator.
>
> # Response to W2
> [4] setup and ours are different.  Their theory setting has no distinction of covariates and labels (it is in a simple univariate gaussian setting; thus completely different from ours). In empirics, they start with fixed real data, fit model, generate synthetic data from it; then iteratively add new synthetic data generated for next step to all available data at previous step and re-train. As one can see, this leads too much injection of synthetic data in the problem quite early on. Hence, it is unsurprising that they observe model collapse. The setting is really completely different from ours: we work under the fresh data augmentation framework from [1].
>
> # Response to W3
> Thanks for the thoughtful comments, this is not true for ridge regression. As explained in the common official response, the observation you refer to holds only for the minimum-norm interpolator. For ridge regression, neat limiting formulae are essential to characterize the optimal mixing ratio. In Theorem 3.3, we show that the ridge optimal weight depends on $\lambda, \gamma$ and the SNR. We have added Figure 3(a) to demonstrate the $\gamma$ dependence. This clearly demonstrates the effect of dimensionality and the importance of our high-dimensional/limiting framework. Unlike us, [5] study low-dimensional GLMs, which show no dimensionality effects, and consider only the MLE. Thus, our entire contribution on ridge theory in high dimensions is new in the model collapse literature. Moreover, among our newly added results, even the optimal weight for min-norm interpolation in case of the ``Time-varying covariates'' setting of Section 5 depends heavily on the dimension. Figure 4(a) (Appendix E) demonstrates this.
>
> We have addressed the minor comment.
>
> # Response to Q1
> See our response to Weakness comment 3 and also our common official comment where we explain the necessity of studying the high-dimensional setting for ridge regression and also for min-norm interpolation in the newly added ``Time-varying covariates'' part of Section 5.
>
> # Q2
> This is a very interesting question. We have added results in this setting under ``Time-varying covariates with fresh labels in a fraction''. See common official comment  for details. In summary, we prove there is a regime of $\gamma$ values where the risk diverges and therefore, model collapse is inevitable. It would be interesting to study the other regime of $\gamma$ values in future.
>
> # Q3
> We have expanded the introduction to elaborate on empirical work in prior literature. The closest empirical work to our paper is done in [1], where they consider fresh data augmentation for training DDPMs on images. Our theory corroborates their finding that model collapse can be prevented in this setting, and we provide strategies for improving the original estimator using synthetic data.
>
> # Q4
> For ridge regression, results under isotropic covariance do not imply results under general anisotropic covariances. Since this is the case, we assume the reviewer meant that for bounded spectra, one can study an equivalent Gaussian sequence model instead. Studying the spectra decay to 0 would be interesting, but we believe should be a different paper, especially given our extensive additions in  Section 5 and Appendix E.
>
> [1] Alemohammad et al. "Self-consuming generative models go mad."
>
> [2] Li et al "Online Forgetting Process for Linear Regression Models."
>
> [3] Gneiting et al. "Calibrated probabilistic forecasting using ensemble model output statistics and minimum CRPS estimation."
>
> [4] Shumailov et al. "The curse of recursion: Training on generated data makes models forget."
>
> [5] He et al. "Golden Ratio Weighting Prevents Model Collapse."

---

> > ### Comment · Reviewer_Fx5M · 2025-11-25
> > **Response to the revision**
> >
> > Thank you for the thoughtful responses, and I really appreciate the revised manuscript addressing my concerns on limited fresh data and varying covariates. I will raise my score accordingly.

---

### Official Review · Reviewer_EFFX · 2025-10-31

**Soundness:** 3
**Presentation:** 3
**Contribution:** 3
**Rating:** 8
**Confidence:** 3

**Summary:**

This paper studies model collapse – i.e. when generative models experience performance degradation when repeatedly trained on its own synthetic data – in overparameterized linear regression for iterative training with a mixture of fresh real labels and synthetic labels. For min-l2-norm and ridge regression, the authors derive asymptotically precise generalization errors and uncover meaningful insights. Core results include: (i) for min-norm interpolation, the optimal mixing ratio equals 1/ golden ratio for general covariance ${\Sigma}$ under certain assumptions; (ii) for ridge, the formula is log-convex in the mixing ratio w, and the optimal ratio is an increasing function of SNR. The paper also treats two case studies on random effect & spiked covariance models and validates theory with simulations.

**Strengths:**

* The precise risk formulae are clean and leverage tools from random matrix theory (beyond isotropic features for the ridge model). The optimal mixing ratio is a meaningful extension from prior works on OLS and low-dimensional results.
* Though technical at times, the paper is written clearly and easy to follow. It shows precise formulae for both min-l2-norm and ridge, including random effect and spiked covariance as examples for the latter.
* The empirical results validate the theory well and show robustness even when certain assumptions are violated.

**Weaknesses:**

I do not identify key weaknesses, and overall the paper is technically sound. There are some small concerns I have:

* **Modeling Gap to Practical Model Collapse**: In iterative training, the synthetic labels are generated from the previous linear weight on the same covariates, but in practice, the model collapse setting touches both features and labels. The bridge from this linear supervised setting to practice seems under-argued. While I appreciate the need for tractable theory, I would like to see some discussions regarding this.

* **Dynamic Mixing**: The idea of dynamic mixing in Section 3.1 is an interesting extension. Though the end result is somewhat similar, I believe it is worth having some synthetic simulations on this too. Additionally, would it be possible to verify these empirical observations on any small-scale experiments with real data?

* **Notations**: at times, vectors and matrices are not bold (e.g. to name a few, Lines 404, 412, 419). It is good to keep the notations consistent.

**Questions:**

Some questions are raised in the Weakness section. Additionally,

1. In the interpolation risk (Thm 3.1), the mixing ratio only affects the variance, but it affects both for the ridge model (Thm 3.2). This is an intriguing phenomenon (something similar also happened in double descent literature as I recall). Do you think there is any possible interpretation for this?

2. How can the results possibly relate to the scenarios using real data?

---

> ### Author Response · Authors · 2025-11-20
> **Response to Reviewer EFFX**
>
> # Response to weakness (Modeling Gap to Practical Model Collapse):
> We appreciate your comment. We have added three new settings in Section 5 (this section is entirely new along with the associated appendix E) that address this. See the official comment above where we describe these additions in details.
>
> # Response to weakness (Dynamic mixing):
> Thank you for the comment. We have added Figure 3(b) to illustrate this phenomenon. It is in fact really interesting. We observe that the optimal weight and the dynamic weight coincide from pretty early on -- in fact beyond the first time point, there is almost no difference. The difference in the first time point may be hard to observe since the plot is quite small in lieu of space but there is actually a difference compared to the remaining time points where they really overlay each other. For instance, at $t=1$, the risk under $\varphi^{-1}$ shows a $0.6$ percent difference relative to risk under dynamic mixing, whereas for higher $t$ the relative average difference is around $0.09$ percent. In lieu of the extensive revision we performed in Section 5 and Appendix E to address other major comments, we defer real data experiments to future, but it would certainly be very interesting to pursue further.
>
> # Response to weakness (Notations):
> We have made these consistent now.
>
> # Response to Q1:
> This is a very interesting comment. Intuitively, there are two sources of bias: 1) from the fact that this is a high dimensional problem, and 2) due to the ridge regularization. Note that the second source of bias affects only the synthetic samples (since these are generated from biased $\hat\beta$'s) and not the real samples. So in ridge regression, the two samples (real and synthethic) have different biases from the regularization, but the same level of bias from the high dimensionality, which is why the weighing appears. However, in  case of the min-norm interpolator the  second source of the bias vanishes. Since now the bias is the same in both real and synthetic samples, we do not see the mixing ratio anymore in the bias term.
>
> # Response to Q2:
> We expect the results would be useful in settings such as [1] (where data needs to be deleted due to govt. regulation) and [2] (where there can be some drift over iterations so previous real labels should not be re-used). As mentioned earlier, in lieu of the extensive revision already performed (see Section 5 and Appendix E), we do not pursue additional real data experiments.
>
>
> [1] Li, Yuantong, Chi-Hua Wang, and Guang Cheng. "Online Forgetting Process for Linear Regression Models." International Conference on Artificial Intelligence and Statistics. PMLR, 2021.
>
> [2] Gneiting, Tilmann, et al. "Calibrated probabilistic forecasting using ensemble model output statistics and minimum CRPS estimation." Monthly weather review 133.5 (2005): 1098-1118.

---

> > ### Comment · Reviewer_EFFX · 2025-11-23
> >
> > Thank you for the response. The answers were insightful and addressed my minor concerns. Since my original score is already high, I will keep my judgment as is. Good luck!

---

### Official Review · Reviewer_8T4F · 2025-11-01

**Soundness:** 4
**Presentation:** 3
**Contribution:** 2
**Rating:** 6
**Confidence:** 4

**Summary:**

This paper studies model collapse in the context of overparameterized linear regression. The authors analyze the minimum-$\ell_2$₂-norm interpolator and ridge regression, in a setting where each iteration mixes real and synthetic labels with a fixed proportion.
The paper provides precise asymptotic risk characterizations using random matrix theory and discuss the dependence on the mixing proportion and the interplay with regularization and SNR. They validate theory with simulations.

**Strengths:**

The paper provides a novel and technically solid contribution to the study of model collapse. It builds on a rigorous mathematical foundation, leveraging random matrix theory to generalize prior results that were limited to low-dimensional or Gaussian settings. The presentation is clear and well-structured, offering useful intuition for interpreting the theoretical findings, which could inform strategies for mitigating model collapse, when fresh labels are available. The numerical experiments are extensively detailed and effectively complement the theoretical analysis.

**Weaknesses:**

The proposed framework may appear somewhat artificial, and its motivation is not sufficiently discussed in the introduction. This limits the perceived practical scope of the work and raises questions about its connection to real-world training dynamics. See also the Questions section.

**Questions:**

1. Could the authors better justify the choice of this framework from a practical point of view? In particular, are there real-world scenarios that resemble the training procedure illustrated here? For instance, how should one interpret the assumption that at iteration $t$, the statistician looses access to the true labels $y_{t-1}, ..., y_1, y$, yet maintains constant access to new labels for the same covariates?
2. Do you have intuition on how would the results change in a setting where, at iteration $t$, the dataset is obtained by union of the fresh and synthetic labels rather than their mixing? Would this alternative formulation still prevent model collapse?
3. As a suggestion, the paper would benefit from a more extended discussion of prior literature. While several recent works on model collapse are cited, a short summary of their different frameworks and main findings would help clarify how the present contribution fits within the literature and emphasize its novelty.

---

> ### Author Response · Authors · 2025-11-20
> **Response to Reviewer 8T4F**
>
> # Response to Weakness:
> We thank the reviewer for their feedback. To further motivate our setting, we have updated the literature review section of our manuscript. Specifically, in the first paragraph, we have expanded on prior observed empirical takeaways from the model collapse literature.
> In Pg. 2 we have added our motivation clearly: essentially, our work is motivated by findings  in [1], which suggest that among interesting possible combinations of real and synthetic data mixing, fresh data augmentation (the setting studied in the original version of our paper) is the only scenario where model collapse can be prevented. Our goal is more to study how model collapse can be prevented in settings where it can. Additionally, to address your comments, we have expanded our results to three new settings in Section 5 with proofs and figure illustrations in the associated newly added Appendix E. Please see the general official comment above for details on these additions. We believe that with these additions your concerns have been addressed.
>
> # Response to Q1:
> Indeed there can be real-world scenarios where it is necessary to forget the previous labels. This can occur specifically in settings where data is observed but cannot be stored or needs to be deleted due to legal regulations [2] or the previous data becomes useless due to drifts in the coefficients of the underlying model [3]. But as mentioned above, if our earlier setting is unsatisfactory, we have added three new settings in Section 5. The first one does not assume fresh labels are available repeatedly  but there our Theorem 5.1 demonstrates that using the synthetic data does not necessarily improve performance (this is in accordance with observations seen in the empirical literature, for instance in [1]). The third setting in Section 5 assumes that fresh real labels are available only on a fraction of covariates. We demonstrate here that the risk diverges with time (see Appendix E.3, Theorem E.2). Both these theoretical results corroborate with empirical observations in the literature, for instance, in [1] that the fresh real data augmentation setting (our original setting) indeed ensures no model collapse. The second setting in Section 5 also has no model collapse -- this is the same as our fresh data augmentation setting but we allow to change the covariates at each iteration.
>
> # Response to Q2:
> If we only use concatenation or union of the datasets with fresh and synthetic labels (instead of weighting the labels), and change nothing else to the setting, this is the same as taking the weight to be half for each of real and synthetic labels in our current setting. So this prevents model collapse but performs suboptimal. In the third setting of Section 5, we have now added results on a generalization of this framework where at each iteration, a fraction of the covariates see fresh labels (also allowing the covariates to change at each iteration). In this case, there are $\gamma$ values where model collapse is inevitable (See Theorem E.2 in Appendix E.3).
>
> # Response to Q3:
> Thank you for this suggestion, we have expanded the introduction (to the extent space allows) to elaborate on contributions in  previous work and contextualize our work better.
>
>
> [1] Alemohammad, Sina, et al. "Self-consuming generative models go mad." The Twelfth International Conference on Learning Representations. 2023.
>
> [2] Li, Yuantong, Chi-Hua Wang, and Guang Cheng. "Online Forgetting Process for Linear Regression Models." International Conference on Artificial Intelligence and Statistics. PMLR, 2021.
>
> [3] Gneiting, Tilmann, et al. "Calibrated probabilistic forecasting using ensemble model output statistics and minimum CRPS estimation." Monthly weather review 133.5 (2005): 1098-1118.

---

### Author Response · Authors · 2025-11-20
**General comment describing major changes**

We thank the reviewing team for the thorough points. The manuscript has undergone major revision with addition of the entirely new Section 5 and the associated Appendix E. The major changes are summarized below. Throughout the below discussion, recall that $\gamma= \lim p/n$.

# (A) Addition of new settings and analyses:
In Section 5, we have included analyses under three different settings: (i) no fresh real data each iteration, (ii) time-varying covariates, that is, covariate matrix no longer stays fixed throughout iterations, and (iii) time-varying covariates with real labels observed on a fraction of them each iteration. Across these, we have addressed effectively all comments raised by reviewers on potential other settings that could be interesting to study beyond the fresh real data augmentation setting we study here. For these additions, we focus on behavior of the min-norm interpolator.

For setting (i), we establish Theorem 5.1, which shows that when fresh real data is not available at each iteration, and only available once at the beginning, the risk is minimized for $w^\star=1$. This means there is no gain from repeatedly adding synthetic data generated from the fitted models. We illustrate this in the newly added Figure 3(c). The details for this setting can be found in Appendix E.1.

For settings (ii) and (iii), we describe our new theorems in the appendix, due to space constraints. In (ii), we allow fresh covariates to be observed at each iteration, keeping all else same. We provide a characterization of the risk in Theorem E.1 (Appendix E.2) but in a reverse n,t order (we explain in Remark E.1 why this is necessary). For isotropic covariates, we further characterize the weight that uniquely minimizes the limiting risk. We demonstrate behavior of this optimal weight in Figure 4 for various values of the dimension to sample size ratio and the SNR.

Finally, in setting (iii), we consider that fresh covariates are available at each iteration but real labels are observed for only half a fraction of these. For the rest, synthetic labels are generated using the model fitted at previous iteration. Thus at each iteration, we effectively observe a semi-supervised problem. The details of this setting can be found in Appendix E.3. We prove that there are regimes of $\gamma$ values where the risk diverges since the bias actually diverges. Thus, model collapse is inevitable for the characterized ranges of $\gamma$ values. This is illustrated in Figure 5 in the Appendix.

Despite these significant additions, we emphasize that our work here is motivated by findings  in [1], which suggest that among interesting possible combinations of real and synthetic data mixing, fresh data augmentation (the setting studied in the original version of our paper) is the only scenario where model collapse can be prevented. Our goal is more to study how model collapse can be prevented in settings where it can.

# (B) Importance/relevance of studying our high-dimensional framework:
We were also asked about the necessity of our high-dimensional framework or rather the necessity of taking limits. We emphasize that this is absolutely critical for ridge regression, which forms a major chunk of our contribution; for instance, see Sections 3.2-3.4. What the reviewers note that the optimal mixing ratio can be derived from the non-asymptotic results in Eq. (A.5) and Appendix C is incorrect for ridge regression. These results apply only for the min-norm interpolator. For the ridge, taking limits is important for obtaining a neat formula for the risk which can then be minimized to characterize the optimal mixing ratio. To illustrate the dependence of this ratio further on the dimensionality, we have added the new Figure 3(a) where we plot the optimal mixing ratio as a function of $\gamma$ for different choices of positive $\lambda$. The dimensionality effect becomes clear here and illustrates how studying this high-dimensional framework is actually important to understand best mixing strategies for ridge regression. We now emphasize this at the end of Pg. 8.

Furthermore, among the added new settings in Section 5 that focus solely on the min-norm interpolator, taking limits is also critical, particularly in setting (ii) on time-varying covariates. This becomes clear on examining Appendix E.2, Theorem E.1, which clearly shows the dependence of the optimal weight on $\gamma$ and also the newly added Figure 4(a), which demonstrates how this weight changes as a function of $\gamma$.

# (C) Expansion on prior work discussion:
We also appreciated comments on the need to expand the literature review, which we have now done in the introduction.

The remaining additions to the manuscript are described in response to reviewer comments below.

[1] Alemohammad, Sina, et al. "Self-consuming generative models go mad." The Twelfth International Conference on Learning Representations. 2023.

---

### Author Response · Authors · 2025-11-30
**Summary of revision and discussions for the new Area Chair**

We summarize the major reviewer comments and our responses below, followed by a summary of reviewer reactions during the discussion phase.

# Key Revisions

Our initial scores were 6, 8, 4, 4. We prioritized addressing the concerns of reviewers Fx5M and LCBS, who initially assigned scores of 4. Both reviewers subsequently raised their scores during the discussion phase (Fx5M to 6 on Nov 25; LCBS to 8 on Nov 20). To address their concerns, we added a new Section 5 with supporting material in Appendix E. Section 5 extends our original analyses to three entirely new settings:

1. **No fresh real data:** The original data is observed only once, and synthetic data is iteratively drawn from the previously fitted model

2. **Time-varying covariates:** The covariate matrix changes over iterations, unlike our original fixed-covariate setting

3. **Time-varying covariates with partial fresh labels:** Covariates change over time with only half receiving fresh labels in each iteration, compared to our original setting where all samples received fresh labels each time.

Details for these three settings appear in Appendices E.1, E.2, and E.3, respectively. Each setting presents distinct challenges, and we derived entirely new theoretical results to analyze them (see Theorem 5.1 and Theorems E.1 and E.2). These new analyses substantially addressed the major concerns raised by both reviewers, as reflected by the reviewer appreciation of our revisions (see further details below).


# Additionally Addressing Individual Reviewer Comments

Reviewer Fx5M additionally requested: (i) comparisons with Shumailov et al., (ii) clarification of connections with He et al., and (iii) discussion of the importance of our limiting risk formulae.  We either clarified these in the discussion or added new text in the manuscript to elaborate further. The reviewer acknowledged our revisions on Nov 25, commenting: *"Thank you for the thoughtful responses, and I really appreciate the revised manuscript addressing my concerns on limited fresh data and varying covariates. I will raise my score accordingly."* They subsequently raised their score to 6.

Reviewer LCBS also asked clarification regarding specific ways our high-dimensional asymptotic framework is useful—we clarified this extensively in the discussions. Notably, this reviewer indicated in their original review that they would raise their score if we addressed even some of their concerns or fixed typos.  We addressed all of their concerns. The reviewer appreciated our extensive revisions, commenting on Nov 20: *"Most of my concerns have been addressed, and the paper now presents a solid theoretical result,"* and increased their score to 8.

Reviewer EFFX scored us 8 initially and requested experiments on our dynamic mixing setting, which we included in Figure 3(b).

Reviewer 8T4F initially scored our paper at 6 and asked us to: (i) comment on the practicality of our setting, (ii) study a setting where real and synthetic data are concatenated rather than mixing labels, and (iii) expand the prior work discussion in the introduction. We addressed point (i) by providing concrete practical examples in our response. For point (ii), we added the "Time-varying covariates with fresh labels in a fraction" subsection in Section 5, which generalizes the requested setting. We also expanded the introduction to clarify prior work. This reviewer did not acknowledge our revision; we sent a reminder to the PCs and area chairs on Nov 24.


*Our updated scores during the discussion phase were 6, 8, 6, 8 (average: 7) before the discussion closed.* Notably, three of four reviewers explicitly acknowledged that we addressed all of their major comments. We sincerely hope our extensive revisions, the reviewers' positive responses, and the updated scores will be considered during the final decision phase.

---

### Meta-Review · Area_Chair_REE1 · 2026-01-07

**Summary:**

This paper considers the problem of model collapse when repeatedly training a model on new synthetic data generated by the old model. In particular the paper restricts itself to the case of linear regression.

The initial reviews by the reviewers were quite split. We had two reviewers voting to accept. Reviewer EFFX giving it an 8 and Reviewer 8T4F giving it a 6. On the other hand, we had two voting to reject with Reviewers Fx5M and LCBS voting to reject.

The major concern across all of the reviewers was the simplicity of the setting. That is, the reviewers felt the setting was too idealized and hence not a good theoretical model to study. In particular the complaints were

1. The same input data $X$ was used at each iteration

2. At every iteration, we had samples from the true generating model. Since the data $X$ does not change, this means, we have the same responses just with new noise. This is a very strong assumption

Finally, reviewer LCBS also had concerns about the order of certain limits being switched.

**Reviewer Concerns:**

### Addressed Concerns

To address these concerns. The paper adds a new section 5, which adds new model settings.

The first is to not have access to real data again but still use the same data $X$. The second is to have access to real data but have time varying data $X_t$. However, while these are more complex than the original setting. I think the correct setting is having no real data and time varying $X_t$.

These new results are restricted. The results for the time varying case are restricted to the identity covariance. Additionally, the second time varying one (Theorem E.2) is only a very small range of $\gamma$ values.

### Unaddressed Concerns

The concern about the switching of the limits was not addressed.

**Reviewer Scores:**

Given the new setting the reviewers did agree that their concerns had been addressed. Hence I believe both reviewers are likely to have increased their scores.

---

### Decision · Program_Chairs · 2026-01-26

Accept (Poster)